# High yield electrosynthesis of oxygenates from CO using a relay Cu-Ag co-catalyst system

Nannan Meng[1,8], Zhitan Wu[1,2,8], Yanmei Huang [3,8], Jie Zhang[1], Maoxin Chen[1,2], Haibin Ma[1], Hongjiao Li [4] ✉, Shibo Xi [5], Ming Lin [6], Wenya Wu[6], Shuhe Han [7], Yifu Yu [3], Quan-Hong Yang [2], Bin Zhang [3] ✉ & Kian Ping Loh [1,2,7] ✉

As a sustainable alternative to fossil fuel-based manufacture of bulk oxygenates, electrochemical synthesis using CO and $H_2O$ as raw materials at ambient conditions offers immense appeal. However, the upscaling of the electrosynthesis of oxygenates encounters kinetic bottlenecks arising from the competing hydrogen evolution reaction with the selective production of ethylene. Herein, a catalytic relay system that can perform in tandem CO capture, activation, intermediate transfer and enrichment on a Cu-Ag composite catalyst is used for attaining high yield CO-to-oxygenates electrosynthesis at high current densities. The composite catalyst Cu/30Ag (molar ratio of Cu to Ag is 7:3) enables high efficiency CO-to-oxygenates conversion, attaining a maximum partial current density for oxygenates of 800 mA cm$^{-2}$ at an applied current density of 1200 mA cm$^{-2}$, and with 67 % selectivity. The ability to finely control the production of ethylene and oxygenates highlights the principle of efficient catalyst design based on the relay mechanism.

Ethanol and acetate acid are essential oxygenates widely used in the industrial manufacture of medicines, polymers, solvents, and daily beverages. The global market size of oxygenates has reached multi-billion USD (for example, USD 89.1 and 20.6 billion for ethanol and acetate acid, respectively) and is anticipated to witness a compound annual growth rate (CAGR) of ~4.8 % from 2022 to 2027 (refs. 1,2). At present, the industrial manufacture of oxygenates relies on grain-based fermentation and fossil fuel-based thermal reactions. The current global food and energy crises, compounded by harsher environmental problems, demand a sustainable production mode to replace traditional methods.

The electroconversion of carbon dioxide/carbon monoxide ($CO_{(2)}$-EC) to oxygenates in aqueous solution at ambient conditions offers a promising solution for transforming trash to treasure, while simultaneously achieving artificial carbon-negative production[3–9]. Compared with present-day $CO_2$-EC, CO-EC addresses the pernicious problems of carbon loss (Formulas 1–4) and instability issues due to carbonate salts in the commonly-used alkaline electrolytes (pH > 7) owing to its non-salifying property[10–12]. Although strong acid electrolytes have been used in $CO_2$-EC, the best performance is often limited to <50% Faradaic efficiency (FE) for $CO_2$-to-oxygenates electroconversion. In addition, the acidity of the electrolyte will continuously

[1]Department of Chemistry, National University of Singapore, 3 Science Drive 3, Singapore 117543, Singapore. [2]Joint School of National University of Singapore and Tianjin University, International Campus of Tianjin University, Binhai New City, Fuzhou 350207, China. [3]Institute of Molecular Plus, Department of Chemistry, Tianjin University, Tianjin 300072, China. [4]School of Chemical Engineering, Sichuan University, Chengdu 610065 Sichuan, China. [5]Institute of Chemical and Engineering Sciences, Agency of Science Technology and Research, 1 Pesek Road, Jurong Island, Singapore 627833, Singapore. [6]Institute of Materials Research and Engineering, Agency of Science Technology and Research, 2 Fusionopolis Way, #0-03, Imnovis, Singapore 138634, Singapore. [7]Department of Applied Physics, The Hong Kong Polytechnic University, Hung Hom, Kowloon, Hong Kong. [8]These authors contributed equally: Nannan Meng, Zhitan Wu, Yanmei Huang. ✉e-mail: hongjiao.li@scu.edu.cn; bzhang@tju.edu.cn; chmlohkp@nus.edu.sg

decline due to the neutralization reaction arising from hydroxyl anion released from water splitting[13]. In view of all these problems, CO is an alternative substrate worth considering because it is a common exhaust gas in industry[14] and the industrial process for $CO_2$-to-CO electroconversion is well established[15]. Increasingly, the emerging consensus is that CO electrolysis is in line with the two-step pathways of $CO_2$-to-CO and CO-to-$C_{2+}$, and study of CO-EC will further our understanding of the overall $CO_2$-EC mechanism given that CO serves as the primary intermediate for C-C coupling in $CO_2$-EC.

$$2CO_2 + 8e^- + 8H_2O \rightarrow C_2H_4O_2 + 2H_2O + 8OH^- \quad (1)$$

$$8OH^- + 4CO_2 \rightarrow 4H_2O + 4CO_3^{2-} \quad (2)$$

namely, clear 2 $CO_2$ and lose 4 $CO_2$

$$2CO_2 + 12e^- + 12H_2O \rightarrow C_2H_5OH + 3H_2O + 12OH^- \quad (3)$$

$$12OH^- + 6CO_2 \rightarrow 6H_2O + 6CO_3^{2-} \quad (4)$$

namely, clear 2 $CO_2$ and lose 6 $CO_2$

Pure Cu can convert CO to multi-carbon compounds via CO-EC[16,17]. There is a growing interest in enhancing the selectivity of the electrochemical pathway to CO-to-oxygenates over Cu under ambient conditions. For example, improving the amount of grain boundaries and roughness factor was reported to enhance the selectivity of CO-to-oxygenates, attaining 100 % FE over Cu[18,19]. However, the reported partial current densities for oxygenates formation (<0.5 mA cm$^{-2}$) at optimal selectivity are significantly lower than the industrial current density (>100 mA cm$^{-2}$), which hampers industrial scaling. Recently, a selectivity of 70-80% for CO-EC and partial current densities of 120–200 mA cm$^{-2}$ have been reported, representing the state-of-the-art[20-22]. However, the selectivity for their target products decreased to less than 50% when the applied current density was over 400 mA cm$^{-2}$[21,22], owing to the competitive HER and preferred production of ethylene[23-25]. It is critical to balance the kinetics of HER with the selective hydrogenation of CO to achieve high faradaic efficiencies for oxygenates production, especially at ampere-level electrolysis.

CO + $H_2O$ → *COH is the limiting step in CO-EC[26,27]. At high concentration of CO, the active sites on Cu for HER become poisoned by CO if the conversion of CO to *COH is slow, and yet a controlled amount of HER is needed to supply proton source for electrochemical reduction. Since Cu has to perform the dual role of CO absorption and HER, increasing the rate of one at the expense of another will be counter-productive for CO-EC. To improve the reaction rate for CO-to-oxygenates electrosynthesis, one strategy is to incorporate a relay system where catalytic site 1 absorbs CO and converts it to a first stage intermediate *COH that diffuses fast to site 2 for further conversion to oxygenates (Formulas 5,6)[28,29].

$$*CO + H_2O \rightarrow *COH + H_2O \rightarrow *CH_x \rightarrow \text{hydrocarbons} \quad (5)$$

$$*CO + H_2O \rightarrow *COH + *CH_x \rightarrow CH_xCO \rightarrow \text{oxygenates} \quad (6)$$

Utilizing the strong CO adsorption capacity of Ag in the liquid phase, an Ag-Cu nanocomposite catalyst is synthesized where Ag nanoparticles foster a COH*-rich environment around Cu nanoparticles covering it partially, and that Ag/Cu interface acts as a sink for the rapid transfer of intermediates. *Operando* X-ray fine structure spectrometry, in situ flow-cell electrochemical Raman spectrometry and online differential electrochemical mass spectrometry confirmed that COH* intermediate is produced on Ag and then transferred to Cu

for tandem reaction, where the enriched COH* intermediate on Cu greatly enhances the partial current density and selectivity for oxygenates electrosynthesis. Our optimized Cu/30Ag catalyst (mole ratio of Cu to Ag is 7:3) shows an optimized partial current density of 800 mA cm$^{-2}$ at driving current of 1200 mA cm$^{-2}$, achieving a decent FE of 67%. The mechanism is further supported by Molecular Dynamic (MD) simulations.

## Results and discussion
### CO adsorption capacity over Ag in the gas phase and liquid phase
Adsorption is the first step in catalytic reactions. In the 1970s, Gossner and coworkers found that Ag exhibits a strong capacity for CO adsorption in aqueous solutions, in contrast to its activity in the gas phase[30,31]. However, Ag is often regarded as a nearly non-adsorbing metal for CO because of its CO-producing property during $CO_{(2)}$-EC (refs. 20,32). To resolve these contradictions, we first studied the temperature programmed desorption of CO (CO-TPD, −60 °C ~ 185 °C) on Ag, with Cu as a control (Supplementary Fig. 1). Figure 1a illustrates the results obtained from CO-TPD experiments, indicating the absence of CO desorption signal on Ag in the gas phase, while a distinct desorption signal is observed on Cu. To further investigate CO adsorption in the liquid phase, linear sweep voltammetry (LSV) measurements in 1 M KOH under both Ar and CO atmospheres were performed. Upon saturating the electrolyte with CO, the onset potential for the HER exhibits a significant negative shift compared to that in Ar-saturated electrolyte (Fig. 1b), indicating strong CO adsorption on Ag and subsequent inhibition of its HER activity. Moreover, as the Ag content in Cu-Ag composites increases, the LSV current densities decrease at the same applied potential under Ar and CO atmospheres, implying that the presence of Ag can suppress the HER side reaction and influence the overall reaction kinetics (Supplementary Fig. 2). CO stripping test in the electrolyte reveals a distinct peak for CO desorption at 0.33 V versus RHE (Fig. 1c, e). Notably, a positive shift of approximately 60 mV is observed compared with that of Cu (Fig. 1d, f), implying a higher bond energy between CO and Ag in the aqueous phase (Supplementary Fig. 3). DFT calculations verify that the presence of an electric field is key to the capture of CO by Ag in liquid phase (Supplementary Fig. 4). This motivates us to synthesize an Ag-Cu composite, leveraging on the CO capture capability of Ag and the tandem reactions facilitated by Cu.

### Synthesis and characterization of Cu on Ag catalyst
To exploit the CO capture property of Ag in electrolytes, a composite of Ag-Cu nanoclusters with varying mole ratios of Ag to Cu is synthesized to investigate their synergistic interactions. According to the standard electrode potentials of Ag and Cu ($E^\ominus_{Ag/Ag+}$ = 0.80 V, $E^\ominus_{Cu/Cu2+}$ = 0.34 V)[33,34], Ag$^+$ is easier to reduce than Cu$^{2+}$. Using ammonia as the reaction medium, the reduction of Ag$^+$ and Cu$^{2+}$ with NaBH$_4$ can be carried out under mild conditions, facilitating the sequential deposition of Cu onto Ag. Using this method, Cu-Ag composite catalysts of different mole ratios (Cu:Ag = 100:0; 90:10; 70:30; 50:50; 10:90; 0:100, corresponding to Cu, Cu/10Ag, Cu/30Ag, Cu/50Ag, Cu/90Ag and Ag for short.) were synthesized (Supplementary Fig. 5). X-ray diffraction (XRD) patterns of the Cu-Ag composites (Fig. 2a) reveal distinct peaks corresponding to Ag (PDF 87-0717) and Cu (PDF 85-1326)[35,36], suggesting the purity of the composites. The increasing Ag-to-Cu content in these composites can be judged from the increased intensity of the Ag (200) peak at ~44.2° and the simultaneous decrease of the Cu (111) peak at 43.3°. Notably, no peak shift is observed, suggesting that Ag and Cu nanoclusters remain as separate entities within the composites[37]. A small diffraction peak at ~36.41° corresponding to $Cu_2O$ (PDF 78-2076) is observed in the Cu, attributed to partial oxidation in air[38]. We focus our discussion on the champion catalyst Cu/30Ag (Supplementary Fig. 6). Figure 2b shows

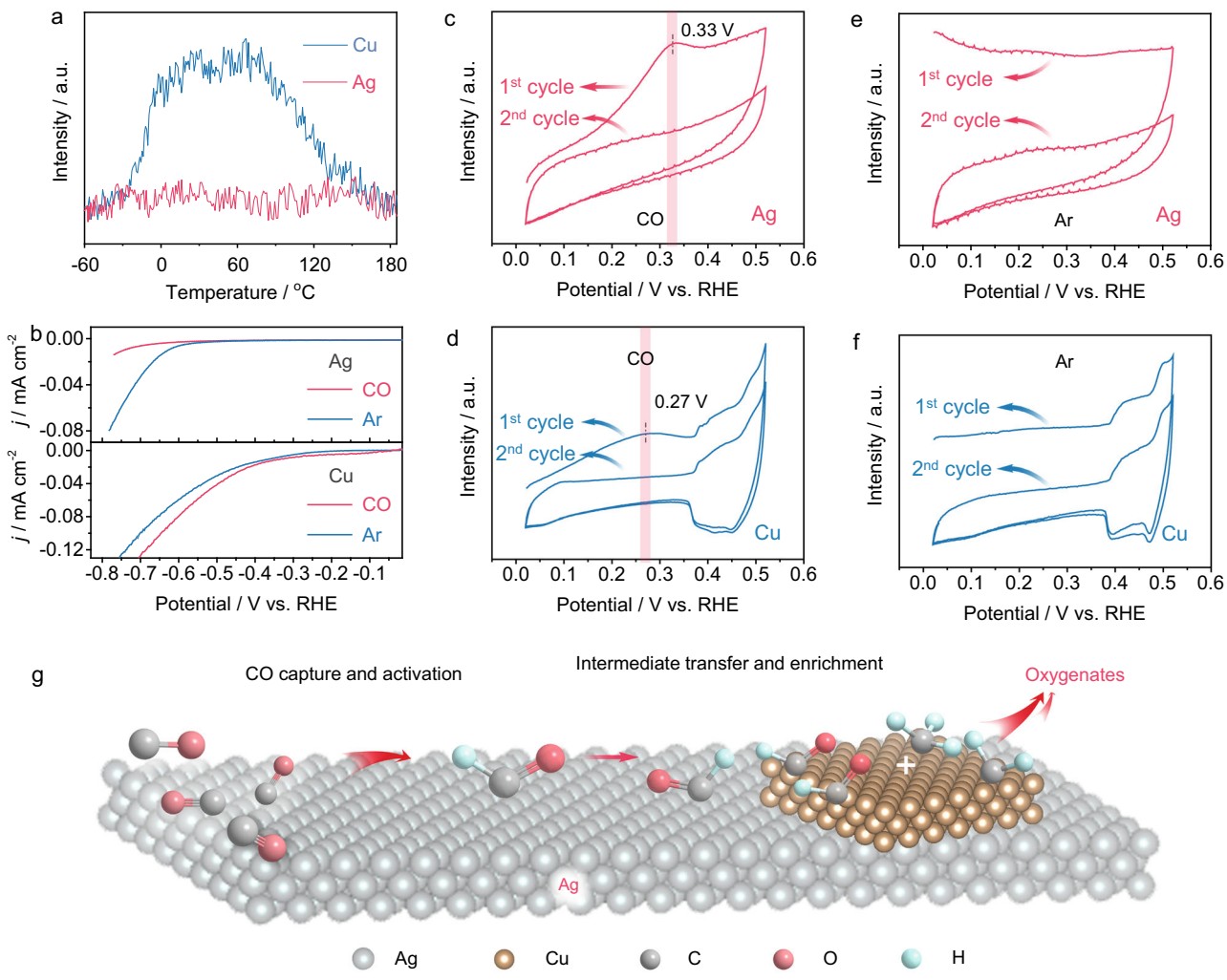

**Fig. 1 | CO capture capacity over Ag in the gas phase and liquid phase. a** Low-temperature TPD curves of CO over Ag (red color) and Cu (cyan color). **b** Linear sweep voltammetry curves of Ag and Cu in 1 M KOH under CO and Ar. **c** CO stripping curve and (**e**) Ar stripping curve over Ag. **d** CO stripping curve and **f** Ar stripping curve over Cu. **g** Scheme describing the tandem reactions on Cu-Ag composite.

its high-angle annular dark-field scanning transmission electron microscopy (STEM) image, along with the corresponding energy dispersive spectra. A line scan analysis of the energy dispersive spectrum from point A to point B (Fig. 2c) confirms the presence of Cu nanoclusters coated onto Ag. High-resolution transmission electron microscopy images (Fig. 2d), inverse fast Fourier transform (FFT) images of selected areas (white and red rectangles in Fig. 2d), and lattice space measurements (Fig. 2e) provide further evidence that Cu clusters with (111) facets are grafted onto the (111) planes of the Ag nanocrystals[39–42]. These results are consistent with the picture of Ag partially covered with Cu nanoclusters, thereby generating an Ag-Cu interface, as well as free Ag and Cu sites. Subsequent sections will demonstrate that all these components play active roles in CO-EC.

## CO-to-oxygenates electroconversion performance over the Cu/Ag electrocatalyst

To assess the catalytic performance of CO-EC, a three-electrode flow cell with an anion exchange membrane was employed in 1 M KOH solution (pH = 14). The chronopotentiometry electroreduction mode was operated in a current density range from 50 to 800 mA cm⁻². The resulting gas products, volatile liquid products, and acetate anion were quantified using gas chromatography, volatile-sensitive headspace gas chromatography and liquid chromatography

(Supplementary Figs. 7–9)[43], respectively. To enhance the mass transfer of CO, a gas diffusion electrode (Supplementary Fig. 10) is employed as the catalyst substrate. Product analysis of pure Cu and Cu-Ag composites (Supplementary Fig. 11, Supplementary Tables 1–5) reveals that acetate salt, ethanol, n-propanol, ethylene and hydrogen from water splitting are the primary products during CO electro-reduction. Minor products include methanol, 1-butanol, ally alcohol, acetone, propionaldehyde, and acetaldehyde.

The faradaic efficiency (FEs) for oxygenates production is typically ~35 % on Cu catalyst, with ethylene as the predominant $C_{2+}$ product, consistent with previous study[22]. For all Cu-Ag composites, ethanol and acetate anion are the dominant oxygenates (Supplementary Fig. 11), suggesting a shared precursor for their formation[29,44]. The FEs for oxygenates show a volcano-like curve due to kinetic limitation imposed by the HER (Fig. 3a). Notably, the FE for oxygenates on Cu/30Ag reaches a maximum value of 60 % at 800 mA cm⁻² while pure Ag shows around 1 % FE for oxygenates (Supplementary Figs. 12, 13 and Table 6). Additionally, the partial current density for oxygenates on Cu/30Ag shows a sustained increase as the total current density increases (Fig. 3b), implying the absence of mass transfer limitation within this range[45]. Reducing the CO flow rate to 5 standard cubic centimeters per minute (SCCM), the single-pass utilization of CO and FE for oxygenates are ~55 % and 56 % at 800 mA cm⁻²

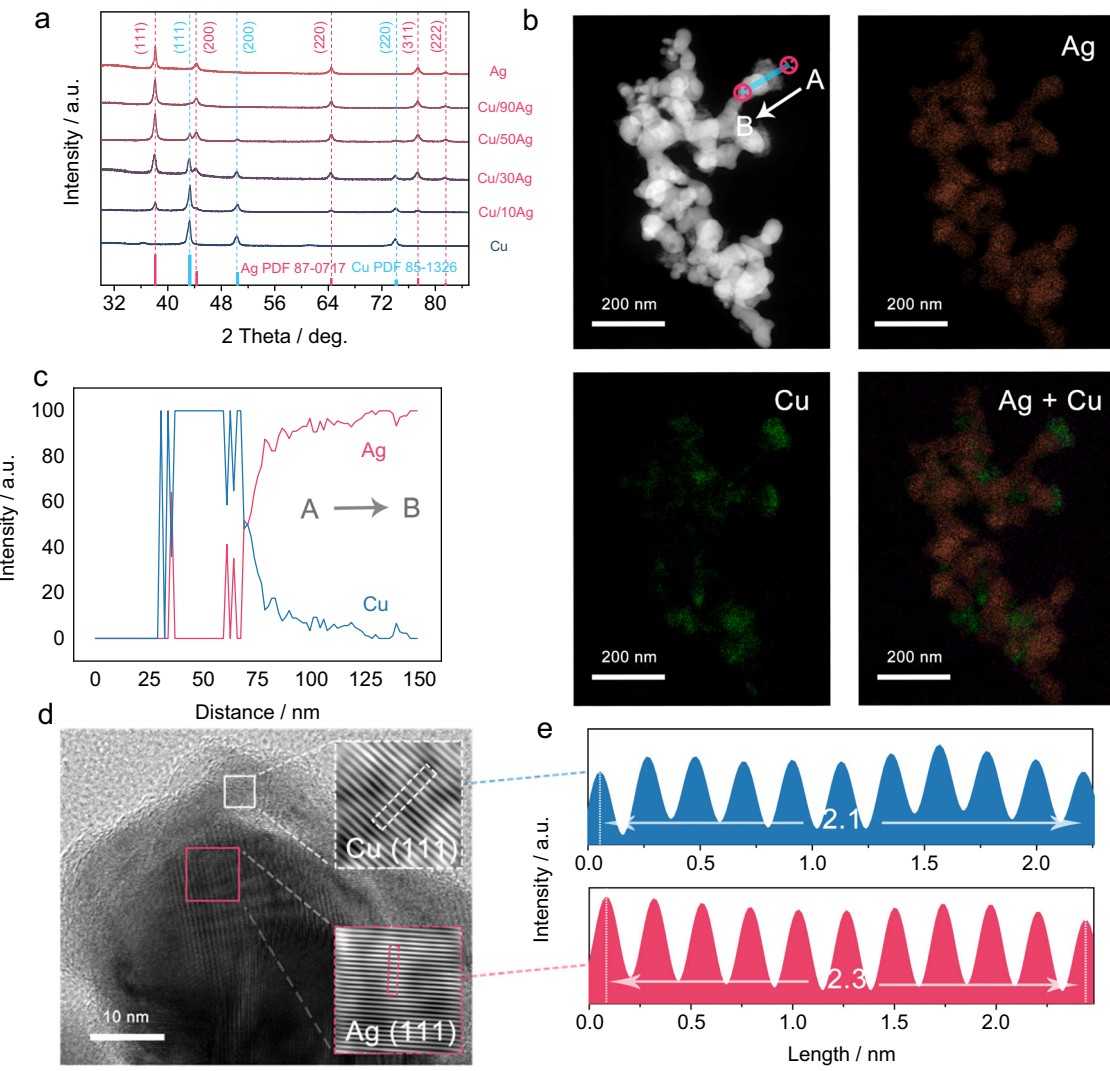

**Fig. 2 | Cu-Ag composite catalyst structure characterization. a** X-ray diffraction pattern of Cu-Ag composites with different composition ratios. **b** A typical high-angle annular dark-field scanning transmission electron microscopy image and the corresponding energy dispersive spectra over the Cu/30Ag composite. **c** The energy dispersive spectrum line scan from point A to point B marked in **b**. **d** High-resolution transmission electron microscopy image and inverse fast Fourier transform images of the selected areas. **e** The lattice space measurement for the white and red rectangles in **d**.

(Fig. 3c, Supplementary Table 7), respectively. This yields an approximate profit of 122 USD per tonne of the potassium acetate product according to techno-economic analysis (TEA) methods used by previous researchers (Fig. 3d)[46,47]. (see Supplementary Note 1 for details).

We discuss the possible role of Ag in the composites. Figure 3e demonstrates that the Ag-to-Cu ratio determines the optimized applied current density for the production of oxygenates, suggesting that the quantity of Ag directly influences the kinetic activity of Cu (Supplementary Fig. 14). Furthermore, the ratios of $FE_{oxygenates}/FE_{ethylene}$ and $FE_{oxygenates}/FE_{hydrogen}$ at their maximum values are calculated using these ratios of Cu at 400 mA cm$^{-2}$ as a reference (Fig. 3f). The $FE_{oxygenates}/FE_{ethylene}$ of all Cu-Ag composites are higher than that of Cu, with Cu/30Ag exhibiting the highest value. The Ag component within the composite serves dual functions: (1) Steering the CO electroreduction product from gaseous ethylene to oxygenates; (2) Suppressing the HER. The performance of the composite is highly sensitive to the Cu:Ag ratio. For example, in case where Ag is excessive relative to Cu, as exemplified by Cu/90Ag, the FE initially increases sharply but reaches a peak at a current density of 100 mA cm$^{-2}$ before decreasing (Fig. 3a). This behavior reflects a mismatch between the generation rate of the intermediate and its consumption rate[28]. After the reaction

at the extreme point, the Cu and Cu-Ag composites show a stable phase structure and morphology similar to their fresh counterparts (Supplementary Fig. 15)

When testing the performance limitations of Cu/30Ag at a high current density of 1.0 A cm$^{-2}$, the FE quickly drops to 36 % owing to mass transfer limitations (Supplementary Fig. 16, Supplementary Table 8). However, by increasing the cathode electrolyte rate to enhance mass transfer, the FE can be recovered and further improved to 61 %, corresponding to a partial current density of 610 mA cm$^{-2}$. The peak performance attained is a FE of 67 % and a partial current density of 800 mA cm$^{-2}$ for oxygenates, with acetate anion and ethanol as the main products (413 mA cm$^{-2}$ for acetate anion and 294 mA cm$^{-2}$ for ethanol.), and 88 % FE for C$_{2+}$ at the applied current density of 1200 mA cm$^{-2}$ (Fig. 3g and Supplementary Table 9). Compared with previous reports on CO electrolysis at room temperature (25 °C) and pressure[20–23,45,48], Cu/30Ag shows one of the best partial current densities and impressive FE for oxygenates electrosynthesis (Supplementary Fig. 17, Supplementary Table 10). Moreover, when using a membrane electrode assembly, the Cu/30Ag catalyst shows stable performance for 28 h at the applied current density of 500 mA cm$^{-2}$ (Fig. 3h, Supplementary Figs. 18, 19 and Table 11).

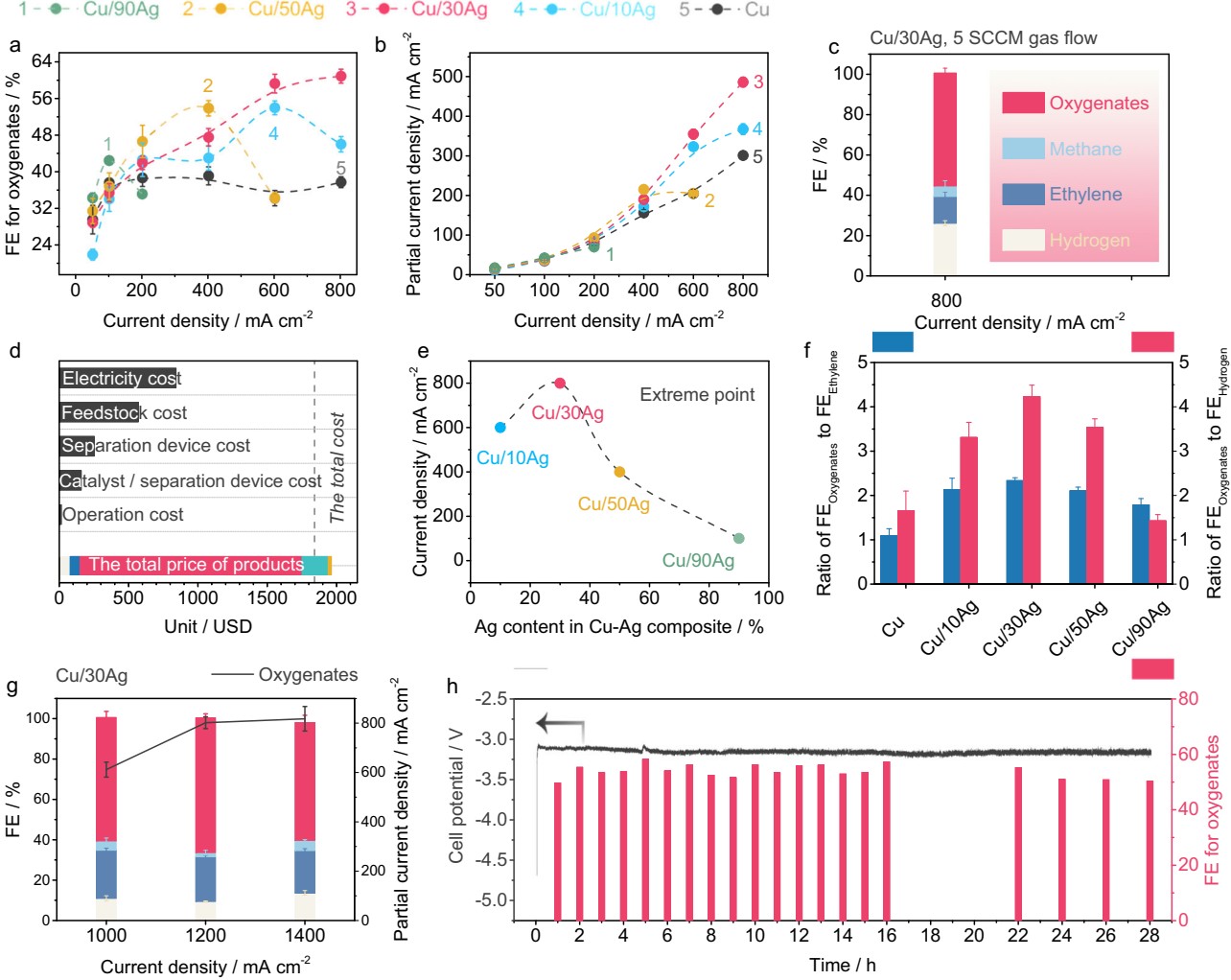

**Fig. 3 | CO-to-oxygenates electroconversion. a** FEs and **b** partial current densities for oxygenates over Cu and Cu-Ag composites under different applied current densities. **c** Cu/30Ag performance for oxygenates electrosynthesis under a CO flow rate of 5 SCCM. **d** TEA analysis. **e** The correlation between the applied current density and Ag content in Cu-Ag composites at their own maximum FE for oxygenates. **f** Ratios of $FE_{Oxygenates}$ to $FE_{Ethylene}$ and ratios of $FE_{Oxygenates}$ to $FE_{Hydrogen}$ over Cu-Ag composites at their own extreme points compared to these ratios of Cu at 400 mA cm⁻². **g** Cu/30Ag CORR performances at the applied current densities of 1000, 1200 and 1400 mA cm⁻² under 100 mL min⁻¹ flow rate of the cathode electrolyte. **h** Stability evaluation of Cu/30Ag using MEA. Error bars correspond to the Standard Deviation (SD) of three independent measurements.

## Electrocatalytic mechanism

*Operando* X-ray absorption near edge structure (XANES) is employed to monitor the chemical state of Cu-Ag during the CORR in real time (Supplementary Fig. 20). As seen from Fig. 4a, the air-oxidized component in Cu/Ag (Supplementary Fig. 21) is rapidly reduced to the metallic state upon the application of a negative voltage (>25 mA cm⁻²)[28,49]. Furthermore, the physically mixed sample synthesized by physical grinding does not match the performance of subsequently deposited Cu/30Ag clusters, suggesting that a chemically bonded Ag-Cu interface is essential for enhancing performance. In fact, the performance of the physical mixture is similar to that of pure Cu (no interface, Fig. 4b) (Supplementary Figs. 22, 23 and Table 12), which is consistent with Yang's work[37]. The areal density of Ag-Cu interfaces as well as the exposed Ag sites could be controlled by adjusting the coverage of Cu on the Ag catalyst. An excessive coverage of Ag by Cu clusters is found to decrease oxygenate production owing to the depletion of surface Ag sites for CO absorption. Conversely, an undercoverage of Ag relative to Cu clusters also results in reduced oxygenate production, as it reduces the density of Ag-Cu interfaces necessary for the transfer of CO intermediates and their subsequent reduction to oxygenates (see description in Supplementary Figs. 24, 25

and Table 13). The partial current density has been normalized against the electrochemical active surface area (ECSA) of the catalyst[19], which is determined by electrochemical capacitance measurement, as well as the roughness factor, and it clearly shows that variation in surface area of the catalysts is not the primary contributors to the CO-to-oxygenate yield (Fig. 4c, Supplementary Figs. 26–28). Instead, the optimal coverage of Cu on Ag, which provides both abundant free Ag sites and Ag/Cu interfaces, plays a crucial role in determining the CO-to-oxygenate yield.

Based on these results, a mechanism is proposed. During the CO-EC, Ag efficiently captures CO (Fig. 1b, c). Local enrichment of CO on Ag sites results in the conversion to oxygenates precursors. Previous experimental and theoretical studies have proved that Ag forms multicarbon products during the CO(2)-EC and even has a lower onset potential for ethanol formation than copper[32,50,51]. Building upon recent works by Carter and Xu's[26–29], we begin with the premise that COH* is the precursor for oxygenates during the CO(2)-EC. Hence, we infer that COH* is produced on the Ag surface. To convert to oxygenates, COH* is transferred from Ag to Cu, whereby COH* will couple with *CH$_x$ generated on Cu[29] during oxygenates electrosynthesis, which not only enhances the production of oxygenates but also blocks

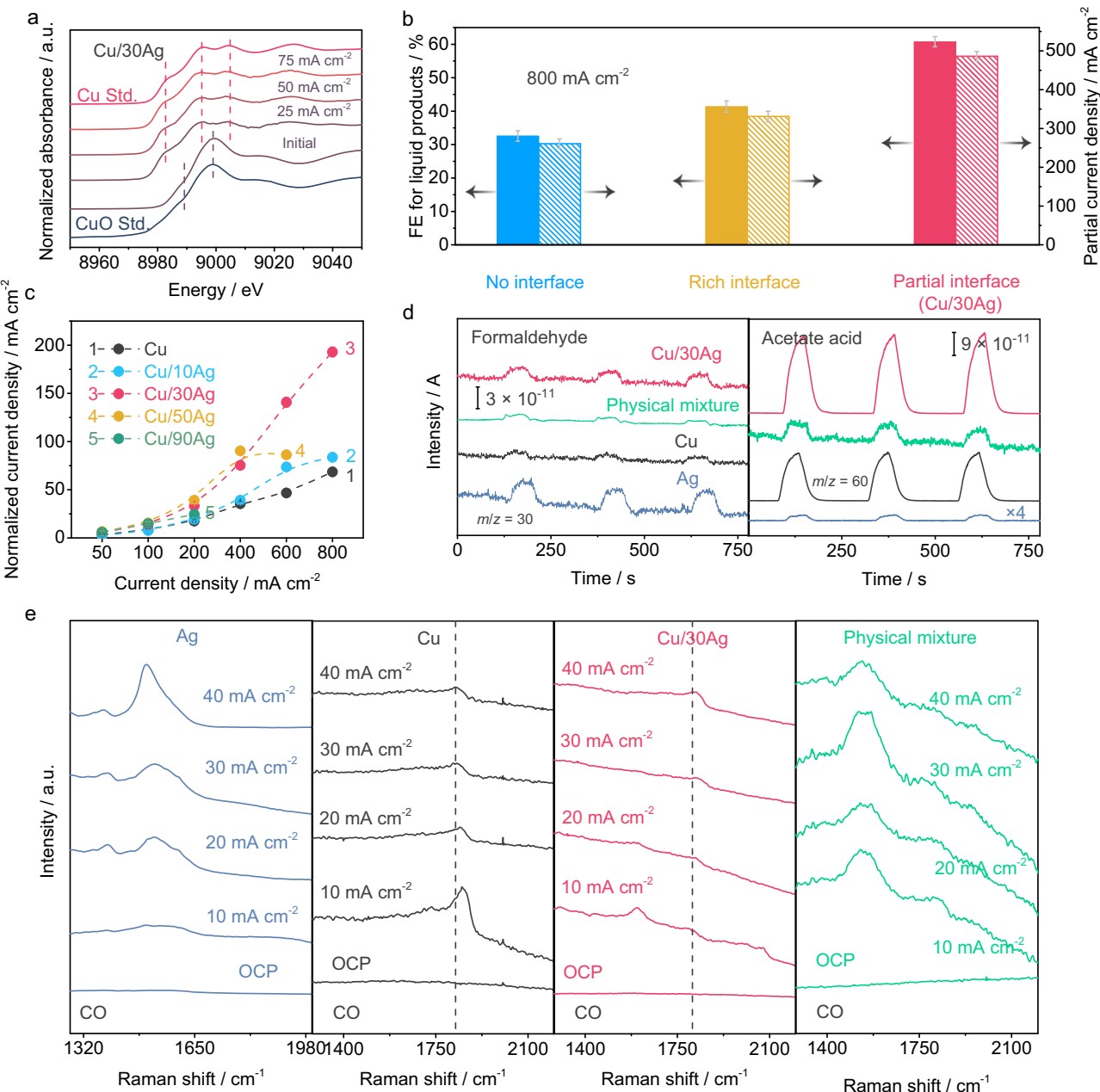

**Fig. 4 | Mechanism exploration for CO-to-oxygenates electroconversion over Cu-Ag. a** *Operando* XANES spectra of Cu in Cu/30Ag during the CORR. **b** The performance comparison of no interface (physically mixed Ag/Cu), rich interface (the sample synthesized by fast reduction in water without ammonia) and partial interface (Cu/30Ag). Error bars correspond to the Standard Deviation (SD) of three independent measurements. **c** ECSA-normalized partial current density of oxygenates. RF was used to replace ECSA for calculation due to a positive relationship between ECSA and RF[19]. Error bars correspond to the Standard Deviation (SD) of three independent measurements. **d** Online DEMS measurements of Cu, Ag, Cu/30Ag and the physical mixture during the CORR. **e** In situ flow-cell-type Raman spectra of Cu, Ag, Cu/30Ag and the physical mixture during the CORR.

the active site for the HER. The enrichment of the COH* intermediate on copper and the suppression of HER help to decrease ethylene generation[24,52]. Finally, the oxygenates production rate will be maximized when the production rate of COH* on Ag and its consumption rate on Cu are matched[28,52]. This mechanism elucidates the necessity for free Ag site in the composite for efficient CO capture and intermediate generation, as well as the importance for interfaces between Cu and Ag for the transfer of intermediate in the tandem reactions. Furthermore, it explains the relationship between oxygenates production selectivity and the existence of an optimal ratio of Cu to Ag in the composite for efficient tandem reaction.

To detect the intermediates generated on the composite Ag/Cu catalyst, online differential electrochemical mass spectrometry (online DEMS) and in situ flow-cell-type Raman cells were used. Online DEMS is an ultrasensitive technique that allows for the detection of volatile species during electrocatalysis[53,54]. To investigate the catalytic mechanism, we first verify whether Ag or Cu alone can generate hydrogenated CO intermediates (e.g., *COH), and then compare these results with Cu/30Ag as well as physically mixed Ag/Cu, which can be considered as separate Cu and Ag nanoparticles without an intimate Ag/Cu intermetallic interface. Here, online DEMS detects the derivatives of *COH such as formaldehyde (HCHO). As shown in Fig. 4d, the

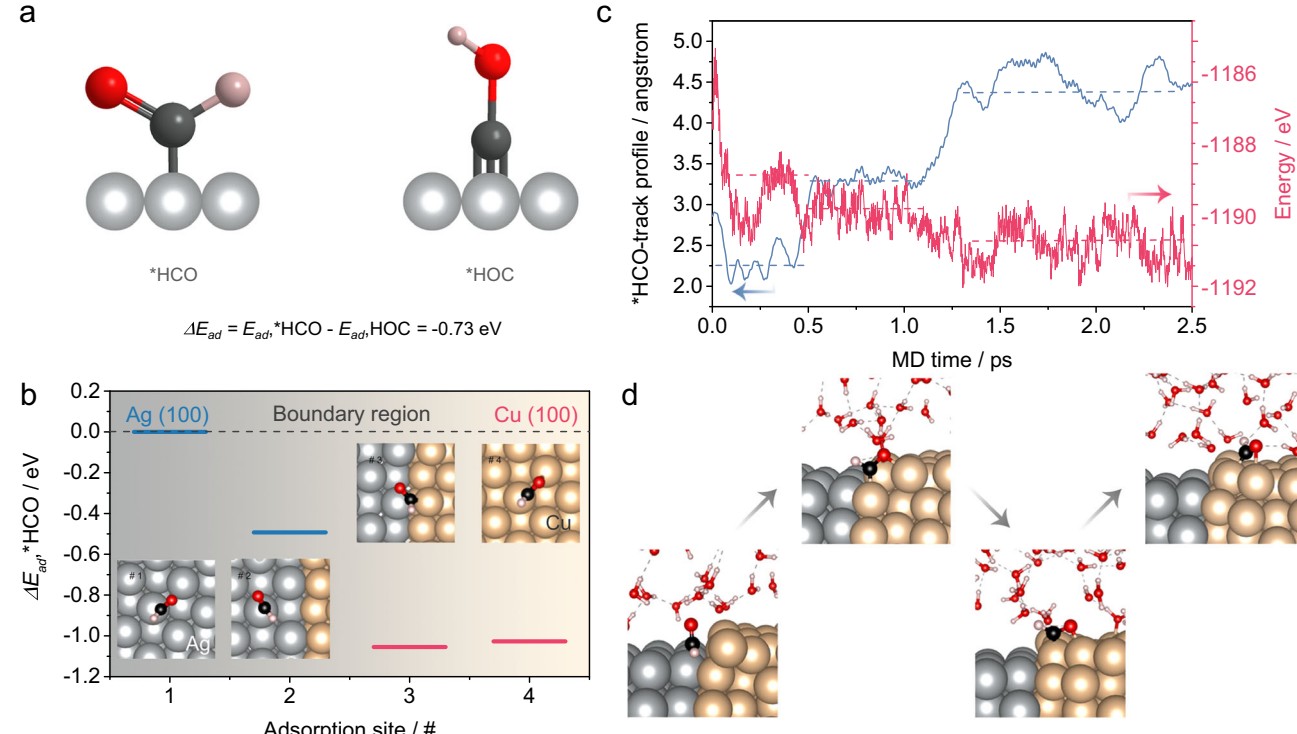

**Fig. 5 | Theoretical understanding of the mechanism. a** Two types of *COH adsorption configurations on the substrate: *HCO and *HOC (red ball, black ball, pink ball and light black ball mean O, C, O and Ag atoms). **b** Changes in the adsorption energy of *HCO from Ag (100) to the boundary region, then to Cu (100)

(inset: top-view snapshots of the dynamic transfer process of *HCO on Ag|Cu(100)). **c** The trajectory profile of *HCO from Ag to Cu and the corresponding energy change. **d** Snapshots of the dynamic process of *HCO on Ag|Cu(100) (red ball, black ball, pink ball, light black ball and yellow ball mean O, C, O, Ag and Cu atoms).

intensity of the formaldehyde signal can be toggled on and off by applying a current in the electrochemical cell, indicating that it is a CO-EC product. Figure 4e clearly shows that pure Ag generates a higher formaldehyde signal than pure Cu, and Cu/30Ag generates higher formaldehyde signal than pure Cu. Meanwhile, the physical mixture of Cu and Ag shows a similar intensity to Cu. In addition, the DEMS signal of Cu/30Ag for acetate acid shows the strongest intensity among pure Cu, pure Ag or physically mixed Cu/Ag samples. The surface enhancement Raman effect of Ag[55] enables the detection of the intermediate species of CORR. As depicted in Fig. 4e, the peak at approximately 1500 cm$^{-1}$, corresponding to HCHO* (ref. [56]), is observed on pure Ag, consistent with the DEMS results. However, the signal is not detected on Cu, as well as Cu/30Ag. The fact that the peak cannot be detected on Cu/30Ag suggests that in the presence of the Cu/Ag interface, the COH* species is transferred rapidly from Ag to Cu and converted to products that rapidly desorb. As a control, an in situ Raman spectrum under Ar gas (Supplementary Fig. 29) is also tested, and no intermediate signal is found, indicating that the signal originates from CO-EC. To confirm that the Cu/Ag interface makes the difference, we also carried out the experiments on physically mixed Cu and Ag. In this case, the COH* species at 1500 cm$^{-1}$ is observed strongly, as would be expected since the presence of Ag in the physical composite plays the same role as pure Ag.

## Mechanistic insights

To elucidate the relay mechanism of *COH on the Ag-Cu catalyst, we constructed a biphase model of Ag(100)|Cu(100) to represent the Ag/Cu interface[51,57]. The monodentate *HCO was found to show a more exothermic adsorption energy than tridentate *HOC on Ag(100) (Fig. 5a). Furthermore, the absorption energy was more exothermic on Cu(100) than on Ag(100), thus providing the thermodynamic driving force for Cu(100) to act as a sink for *HCO activated by Ag(100) after

the capture of CO (Fig. 5b). These trends are also manifested by interface of Ag(111) and Cu(111). Theoretical analysis indicates that hydrogenation of CO is thermodynamically favored over CO dimerization on both Ag and Cu surfaces (Supplementary Fig. 30 and Table 14). To investigate the dynamic evolution of this transfer process in the presence of an electrolyte (mimicked by a water phase), ab initio molecular dynamics (AIMD) simulations were performed (Supplementary video 1, Supplementary Fig. 31). The trajectory profiles of *HCO from Ag to Cu across a Ag-Cu interface clearly revealed three stages of transfer, namely, the adsorption of *HCO on Ag, followed by the interaction of *HCO with the Ag-Cu boundary, and finally, the migration of *HCO to Cu, all occurring within a timescale of 2 ps (Fig. 5c, d). Considering that Cu(111) surface and Ag(111) faces are typically the dominant facets of polycrystalline Cu and Ag, we have also carried out AIMD simulations for the (111) surfaces of these crystals, and the results revealed that similar to the (100) faces, *HCO is spontaneously transferred from Ag(111) to Cu(111) (Supplementary Figs. 32 and 33). Therefore, such a relay mechanism clearly explains the need for high-density Ag-Cu interface for the efficient capture and pre-hydrogenation by the Ag site, followed by transfer and further hydrogenation by Cu site. In addition, the efficient capture and transfer of *HCO by Ag and its fast transfer to Cu suppresses other C2 generation pathways and increases the selectivity for oxygenates.

## Discussion

Leveraging on the strong CO adsorption on Ag in the liquid phase, we designed and constructed Cu-Ag bimetal catalysts featuring abundant Ag sites as well as Ag-Cu interfaces, both of which were shown to be essential for CO capture and activation in tandem. The champion Cu-Ag catalyst achieved a high faradaic efficiency of 67 % for CO-to-oxygenates electrosynthesis with a partial current density of 800 mA cm$^{-2}$ in 1200 mA cm$^{-2}$ electrolysis under ambient conditions.

In situ characterization techniques verified that the COH* intermediate produced on Ag was transferred rapidly to Cu. This enrichment of oxygenate intermediates on Cu not only enhanced oxygenates formation but also suppressed the pathway for ethylene and side HER reactions, which is further supported by theoretical calculations. Insights obtained on the Cu-Ag catalytic relay system in CO electroreduction pave the way for developing other efficient catalysts for the green manufacture of oxygenates.

## Method

### Material
All reagents and consumables were purchased from commercial sources without further purification. $AgNO_3$ (ACS reagent, 99%), $Cu(NO_3)_2 \cdot 3H_2O$ (Puriss, 99%), $NaBH_4$ (98%), KCl (Puriss, 99 %), Nafion$^{TM}$ 117 containing solution (-5% in a mixture of lower aliphatic alcohols and water), KOH (Reagent grade, 90%), acetaldehyde (ACS regent, 99.5%), propionaldehyde (Reagent grade, 97%), 1-butanol (ACS regent, 99.4 %), ally alcohol (99%) acetone (HPLC, 99.9 %), methanol (HPLC, 99.9%), n-propanol (HPLC, 99.9%) and ethanol (HPLC, 99.9%) and acetic acid (HPLC, 99.8%) were purchased from Sigma-Aldrich. $NH_3 \cdot 3H_2O$ (28%) was purchased from VWR, and the Selemion AMVN anion exchange membrane was purchased from Asahi Glass Company. An Ag/AgCl (3 M) reference electrode (diameter 3.8 mm) was purchased from Shanghai Chuxi Industrial Co., Ltd. An AvCarb GDS3250 gas diffusion electrode (GDE) was purchased from Fuel Cell Store.

### Synthesis
The total moles of ($Cu(NO_3)_2$ + $AgNO_3$) was fixed at 6 mmol and dissolved in 70 mL of water. Next, 10 mL of $NH_3$ solution was dropped into the above solution. $NaBH_4$ (0.5 M) solution was added into the above-formed transparent solution until the reaction was complete. The obtained product was washed with water 3 times and then ethanol 3 times. Finally, the wet product was dried at room temperature in a vacuum drying box. By controlling the feed molar ratio of Cu:Ag (100:0, 90:10, 70:30, 50:50, 10:90 and 0:100), Cu-Ag composites (Cu, Cu/10Ag, Cu/30Ag, Cu/50Ag, Cu/90Ag and Ag for short) were obtained. For comparison, a sample with the same component ratio as Cu/30Ag was prepared by directly reducing ($Cu(NO_3)_2$ + $AgNO_3$) in water without $NH_3$ and named interface-rich Cu/30Ag. The physical bimetallic mixture of (Cu + Ag) with the same component ratio as Cu/30Ag was prepared by physical grinding and named physically mixed Cu/30Ag.

### Electrode preparation
The catalyst was added to a mixed solution of Nafion and isopropanol. After 30 min of sonication, the homogenous dispersion was sprayed on the GDE, and the loading amount was 1 mg cm$^{-2}$. Specifically, the catalyst ink was prepared by adding 100 μL of Nafion into 5 mL of isopropanol in which 25 mg of catalyst powder was dispersed, followed by ultrasonication for 30 min. The well-mixed catalyst ink was then spray-coated onto a GDE (5 cm × 5 cm) by airbrushing to achieve a catalyst loading of 1 mg cm$^{-2}$. After drying in a vacuum drying box for 2 h at room temperature, the catalyst-loaded GDE was obtained.

### Characterization
The X-ray diffraction pattern (XRD) of the sample was collected on a Bruker D8 Advance diffractometer using Cu Kα radiation (0.1518 nm). The morphology and phase distribution were analyzed on a JEOL JSM-6701F field emission scanning electron microscope and an FEI Tian transmission electron microscope. The element distribution of the sample was obtained by energy dispersive X-ray elemental mapping and line scan. Scanning transmission electron microscopy analysis was conducted using a JEOL ARM200F microscope operating at 200 kV equipped with a cold field emission gun. The microscope featured an aberration corrector CEOS for atomic resolution.

### *Operando* X-ray absorption near-edge structure spectroscopy test
Operando X-ray absorption near-edge structure spectroscopy of the sample (XANES) was collected on the XAFCA beamline of the Singapore Synchrotron Light Source in the fluorescence pattern model. The operando cell with 1 × 1 cm$^2$ electrode area was homemade and X-ray light hit the catalyst-covered gas diffusion electrode and then reflected to the signal receiver. The cell was sealed with Kapton tape in case of CO gas leaks (Supplementary Fig. 14). XANES data were simultaneously calibrated by recording the transmission spectrum of Cu foil.

### Online differential electrochemical mass spectrometry test
Online DEMS (Shanghai Linglu Instrument & Equipment Co.) is a volatile-sensitive detection technique and can capture tiny amounts of matter during the real-time electrocatalytic process. The signals of matter were acquired though a hydrophobic, gas-permeable polytetrafluoroethylene membrane that separated the aqueous reaction system and nonaqueous detection system. Specially, 2 mg of the catalysts, 20 μL Nafion (5 wt.%) solution, and 980 μL ethanol were mixed and ultrasonicated for 20 min. Then, 100 μL of the ink was dropped on the glassy carbon electrode (-0.71 cm$^2$) and used as the working electrode. Meanwhile, the Hg/HgO electrode (1 M KOH) and platinum wire electrode were put into the cell and acted as counter and reference electrodes, respectively. 1 M KOH was continuously circulated through the electrochemical cell using a peristaltic pump. The volatile products generated ($m/z = 30$ for formaldehyde, $m/z = 60$ for acetate acid) were transported to the mass spectrometer using its inner pump. The detection of products was facilitated by a hydrophobic polytetrafluoroethylene (PTFE) membrane, crucial for permitting the passage of volatile substances while blocking water from entering the vacuum chamber[53].

### In situ flow cell electrochemical Raman spectroscopy test
In situ flow cell Raman spectroscopy was performed on a Horiba LabRAM HR Evolution equipped with a He-Ne laser (633 nm wavelength). An in situ Raman cell was designed by removing both the anode chamber and anion exchange membrane of the CO-EC reaction cell. The reserved cathode chamber and flow gas chamber allowed the Raman probe to insert into the electrolyte and approach the catalyst-covered gas diffusion electrode. Raman signals were collected after carrying out the chronopotentiometry electroreduction mode.

### Temperature programmed desorption of CO test
All catalysts were initially reduced using hydrogen gas at 400 °C for 1 h before the temperature programmed desorption of CO test (Micromeritics, AutoChem1 II 2920). The system was then cooled using liquid nitrogen. Then, the CO/He mixture (10/90, V/V) was flowed into the system until saturated adsorption. Next, He gas flowed into the system. Finally, the temperature of the system was increased at 5 °C min$^{-1}$ for CO desorption, and the data were recorded using mass spectrometry.

### CO stripping test
For testing CO stripping, all catalysts were initially reduced at −1.8 V versus Ag/AgCl for 10 min under an Ar atmosphere in an H cell. The effective membrane area of the H-cell was 4.3 cm$^2$, and the volumes of the cathode and anode electrolytes were both 50 mL. The effective dimension of the electrodes was 1 × 1 cm$^2$. Subsequently, CO gas flowed into the cell for 10 min without electricity. Next, −1.8 V versus Ag/AgCl was carried out for 10 min under a CO atmosphere to capture CO. Then, the CO gas was switched to Ar and flowed into the cell without electricity for 10 min to remove free CO in the cell. Finally, the CO stripping curve was acquired using the cyclic voltammetry method. For comparison, Ar stripping was also tested by the replacement of CO with Ar, and other conditions were kept the same as the procedures of CO stripping. Here, the reversible hydrogen electrode was adopted in

this work unless otherwise mentioned. $E_{RHE} = E_{Ag/AgCl} + E^{\ominus}_{Ag/AgCl} + 0.059 \times pH$, $E^{\ominus}_{Ag/AgCl} = 0.197$ V.

## Electrochemical active surface area test

The electrochemical active surface area was collected in the non-Faradaic potential region using the electrochemical capacitance measurement method. The sample was analyzed under different scanning rates from 20 mV s⁻¹ to 120 mV s⁻¹. The relative electrochemical active surface area was acquired by normalizing the electrochemical capacitance of the sample to that of the gas diffusion electrode.

## Product quantification

Gas products, volatile products and organic salts were quantified using an online gas chromatograph (Agilent, 8890 GC) integrated with a thermal conductivity detector, a flame ionization detector and a head-space gas chromatograph (Agilent, 8890 HSGC) with a flame ionization detector. Liquid salts were analyzed by high-performance liquid chromatography (Agilent, HPLC-1260 Infinity) with an Aminex HPX-87H column (Bio-Rad, 300 × 7.8 mm) using 0.5 mM $H_2SO_4$ as the mobile phase. Faradaic efficiency = $n \times N \times F/I/t \times 100\%$; Here, $n$, $N$, $F$, $I$ and $t$ correspond to the moles of the product, the number of electron transfers, the Faraday constant, the applied current density, and the reaction time, respectively.

## CO electroconversion test

CO electroconversion was performed in a flow cell with a three-electrode system. The working electrode, reference electrode and counter electrode corresponded to the catalyst-covered gas diffusion electrode, Ag/AgCl electrode and Ni foam, respectively. The effective membrane area is 1 × 1 cm². The cathode and anode chambers were separated by a Selemion AMVN anion exchange membrane that protected organic products against overoxidation on the anode side. Before electrolysis, the flow rates of both anodic and cathodic electrolytes (both were 1 M KOH, pH = 14) were set at 10 mL min⁻¹, and the flow rate of CO gas (air liquid, 99.97%) was set at 30 mL min⁻¹ via a mass flow controller (MC-500SCCM-D, ALICAT). The flow rate of gas products was monitored by the other mass flow controller located between the cell and GC. Anodic and cathodic electrolytes were equipped with a pump so that they could form the recycling loop. The effective catalytic area was 1 cm². Finally, chronopotentiometry electroreduction mode (Gamry, Reference 3000) was carried out. For every applied current density, products were quantified after 600 s of electrolysis and at least three replicates were performed to yield an average and the standard deviation. In case of toxic CO leakage, those experiments should be conducted in the fume hood and equipped with a CO senser. Here, no iR correction was performed in this work.

## Theoretical methods

All the DFT and AIMD calculations were carried out with the Vienna Ab initio Simulation Package (VASP 6.0)[58] with the Perdew-Burke-Ernzerhof (PBE) exchange-correlation functional[59] and projector augmented wave (PAW) potentials describing the ionic cores[60]. The Ag|Cu biphase models were first fully relaxed as the bulk phase, which was followed by the relaxation of the slab models (60 Ag atoms + 60 Cu atoms and 48 Ag atoms + 60 Cu atoms) with the bottom two layers fixed at the optimized bulk positions (Supplementary Fig. 34). All the single phase models of Cu(100) and Ag(100) models were employed with (3 × 3) repetitions. For DFT calculations, the vertical vacuum separation of repeated images was more than 15 Å and dipole corrections were applied. The geometry optimizations were performed with a plane-wave cutoff of 400 eV and the convergence was reached when the maximum force on any atom was below 0.05 eV Å⁻¹. For AIMD simulations, the simulation boxes contain metal slabs with adsorbates as well as the water phase (58-60 $H_2O$ moleculars depending on the adsorbates) to mimic the electrolyte|electrode interface. An NVT ensemble with a Nosé–Hoover thermostat at 298.15 K was employed, and the simulation period was 2 ps with a time step of 1 fs. The Brillouin zones of all systems were sampled with Monkhorst–Pack grids[61]. The k-point sampling (k1, k2, k3) was such that the product between its components and the norms of the supercell vectors (a, b, c) was at least (25 Å, 25 Å, 25 Å), which ensures that the meshes are dense enough.

## Data availability

The data that support the plots within this paper are available from the corresponding author. Source Data for the main text figures are provided as a Source Data file. The configuration files with all parameters for AIMD simulations and relaxed geometries through DFT are provided as Supplementary data 1. Source data are provided with this paper.

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

## Acknowledgements

The authors are grateful to the National Natural Science Foundation of China (No. 22109115 (N.M.)), the Research & Experimentation group at

Shell (A-0004543-01-00 (K.L.)) and Jockey Club 2D Quantum Laboratory and Global STEM professorship scheme project (P0043063 (K.L.). We acknowledge the kind support of Rong Yang of Tianjin University.

## Author contributions

K.L. and B.Z. conceived the idea and directed the project. N.M. designed the experiments. N.M., Z.W. and M.C. carried out the materials synthesis, characterization and performance measurements. H.L. contributed to the theoretical calculation. Y.H. contributed to electrochemical online DEMS. J.Z. and H.M. assisted in electrochemical in situ Raman spectroscopy. S.X. assisted in XAFS. M.L., W.W. and S.H. assisted in TEM. N.M., Z.W. and H.L. cowrote the paper. K.L., B.Z., Y.Y., and Q.-H.Y. revised the manuscript. All authors discussed the results and commented on the manuscript.

## Competing interests

The authors declare no competing interests.
