## [Peer Review File · Nature Communications]

High yield electrosynthesis of oxygenates from CO using a relay Cu-Ag co-catalyst systemREVIEWER COMMENTS

Reviewer #1 (Remarks to the Author):

The authors designed Cu-Ag bimetal catalyst with Ag-Cu interfaces, which are crucial for CO-to-oxygenates electrosynthesis, achieving a partial current density of 800 mA cm⁻² of high faradaic efficiency of 67%. Also by in-situ characterization techniques and theoretical methods to confirm the mechanistic pathway, the authors showed impressive results. However, I would like to recommend this manuscript for publication, but only after the authors clarify satisfactorily on two major key points.

The authors have described the association between the *COH intermediate (formaldehyde) and the main product, acetic acid, using online DEMS and in-situ Raman analysis. However, in the Cu/30Ag catalyst configuration, ethylene also occupies a significant portion as a main product. The authors should additionally explain the high selectivity of *COH intermediate towards ethylene.

The authors assumed an Ag(100)/Cu(100) interface for the DFT and AIMD calculations and described their findings based on this assumption. However, when examining the XRD data in Figure 2a, it is evident that the Cu/Ag catalyst does not exhibit the crystallographic orientation of Ag(100) or Cu(100), but rather prominently displays Cu(111) and Ag(111) orientations. Therefore, in order to establish a connection between the DFT and AIMD calculation data and the experimental data, it is necessary to provide calculations for the Ag(111)/Cu(111) interface.

Reviewer #2 (Remarks to the Author):

In this article, the authors investigate Cu-Ag nanocomposites for CO electroreduction to oxygenate products. Their premise is that CO adsorbs more strongly in the liquid phase on Ag than on Cu, contrary to trends in the gas phase. This has been demonstrated using TPD curves in the gas phase, LSV, and stripping experiments in the liquid phase. Therefore, if a composite Cu-Ag catalyst is used, CO adsorbs on Ag, which they claim can convert to COH* and is then transported to Cu, where it undergoes C-C coupling reactions. They used sequential deposition techniques to form the desired nanocomposites with different Cu-Ag ratios. Experimental characterization using XRD, STEM, HR-TEM shows no alloy formation is observed but the Cu nanoclusters coated by Ag. They also observe lattice fringes corresponding to the 111 facets of both metals in TEM. They then performed flow cell experiments (MEA) at high current densities to show that the FE for oxygenates is indeed high at 800

mA/cm², with Cu/30Ag performing the best. The stability of the catalyst for 28 hrs shows no degradation of performance. To understand the mechanism behind the improvement, they performed DEMS and in-situ flow-cell experiments where they observed peaks corresponding to the formation of formaldehyde on Ag while the same was not observed on Cu or Cu/30 Ag. Through this, they conclude that Ag helps in the formation of formaldehyde-like species. However, they claim that on the interface even though formaldehyde like species is produced, they are consumed very fast leading to non-detection of these species. They then performed a few DFT/AIMD simulations to check the stability of COH on interfaces and the movement across it. The article makes interesting observations regarding the role of Ag, and the characterization and catalytic experiments look promising. However, the paper raises a lot of questions, especially in terms of mechanistic aspects and the DFT simulations which do not merit the direct publication of the work.

1. The authors, based on previous articles (ref. 26, 27 in the paper), make an assumption that CO to COH can be rate-determining for C₂ formation instead of C-C coupling steps. Both the references that the authors cite do not definitively conclude that only CO to COH is RDS; they claim that CO to COH/CHO can be RDS for this reaction. However, the authors only consider the COH* formation as RDS, but following their logic, how do they explain the formaldehyde product detection from their DEMS and Raman experiments on Ag? Formaldehyde will be produced from the CHO pathway, not the COH (Energ Environ Sci 2010, 3 (9), 1311–1315). COH converts to carbon or CHOH and then leads to methane formation. I feel this assumption of COH formation as RDS cannot help in understanding or explaining the experimental results.

2. Based on the mechanism proposed, how do authors explain the increased selectivity to oxygenate products? If all the C-C coupling does occur on the Cu surfaces, why is there an increase in oxygenate product formation? Since Cu is inherently known to produce both ethylene and ethanol.

3. The authors should also consider going through the following paper and possibly cite it: Acs Catal 2020, 10 (7), 4059–4069. Even in this work, CuAg composite catalysts were used albeit for CO₂ reduction with more detailed DFT simulations.

4. Furthermore, the current set of DFT simulations in the paper adds minimal value to the study while a lot more can be done.

4(a). Firstly, the basic assumption of using Cu(100)|Ag(100) interface is not at all justified. The HRTEM in Figure 2(d) shows lattice fringes corresponding to Cu(111)|Ag(111). Why wasn't this interface simulated? The current Cu(100)|Ag(100) interface looks very sub-optimal to me; what is the lattice strain of the interfaces when they are placed side-to-side.

4(b). What is the difference in thermodynamic or kinetic barriers for COH/CHO formation from CO on Ag vs. Cu surfaces. Since CO hydrogenation step is claimed to be the RDS in this study and how does it compare to C-C coupling reaction energetics.

4(c). The authors make an interesting claim through experiments in Figure 1 that CO adsorption can actually be stronger on Ag as compared to Cu in a liquid medium. Can this be proved using DFT simulations?

4(d). In the AIMD simulations in Figure 5c, how can authors justify a change in energy of 6 eV? Furthermore, I cannot fathom COH* displacing so much and crossing the interface in just 2 ps of AIMD simulations. I would have expected the authors to use some accelerated AIMD techniques like meta dynamics or slow-growth simulations to simulate this process.

Reviewer #3 (Remarks to the Author):

Herein, Meng et al successfully address a long-standing scholarly debate concerning CO capture and activation over Ag. Furthermore, leveraging this property, they propose a rationally designed strategy to employ Ag as a catalytic site for capturing CO and converting it into a COH* oxygenates precursor. This process creates a COH*-rich environment around Cu nanoclusters, serving as a highly efficient relay electrocatalyst for synthesizing oxygenates products. The effectiveness of Ag and the transfer of COH* from Ag to Cu are solidly demonstrated through CO stripping, in situ DEMS/Raman/XAFS experiments, robustly supported by Ab initio molecular dynamics simulations. By briefly controlling the areal density of Ag-Cu interfaces to match or balance the generation rate of the oxygenates precursor with its consumption rate, the record partial current density of 800 mA cm⁻² with decent 67% selectivity in 1200 mA cm⁻² CO electrolysis is reached. Significantly, through optimization of reaction parameters, this results in an approximate profit of 234 USD per tonne of acetic acid. As a result, this well-organized manuscript offers a novel understanding of the Cu-Ag catalytic system and contributes to enhancing overall electrolysis efficiency for oxygenates electrolysis. Given the significance of these findings and the solid experimental/mechanism analysis, I strongly recommend the publication of this manuscript in Nature Communications. Minor revisions are suggested to address the following issues.

1. For further reflecting the CO capture capacity of Ag in the composite, the CO stripping curve of the champion Cu/30Ag catalyst should be added.
2. The V-t curves over the champion catalyst under the different applied current densities (50 mA cm⁻² to 800 mA cm⁻²) should be provided for further describing its stability.

3. Table of Contents (TOC) of this work can be drawn and provided.
4. To enhance the reproducibility of this study, the additional information regarding the electrode preparation process should be added.
5. Page 10, line 22. '234 USD per tonne of the products' should be revised to '234 USD per tonne of the product' or '234 USD per tonne of the acetate acid product'.

A point-by-point response to the reviewers' and editor's comments

To reviewer 1:

Reviewer letter: The authors designed Cu-Ag bimetal catalyst with Ag-Cu interfaces, which are crucial for CO-to-oxygenates electrosynthesis, achieving a partial current density of 800 mA cm⁻² of high faradaic efficiency of 67%. Also by in-situ characterization techniques and theoretical methods to confirm the mechanistic pathway, the authors showed impressive results. However, I would like to recommend this manuscript for publication, but only after the authors clarify satisfactorily on two major key points.

Answer: We highly appreciate the reviewer for the positive and constructive comments on our communication. To save the reviewer's valuable time, key revisions are displayed in yellow background in the revised manuscript and the revised supporting information.

Comment 1. The authors have described the association between the *COH intermediate (formaldehyde) and the main product, acetic acid, using online DEMS and in-situ Raman analysis. However, in the Cu/30Ag catalyst configuration, ethylene also occupies a significant portion as a main product. The authors should additionally explain the high selectivity of *COH intermediate towards ethylene.

Answer: We sincerely acknowledge the kind comment. According to the reviewer's suggestion, we have now revised and stated that ethylene also occupies a significant portion as a main product and added in the revised manuscript.

Herein, the increased intermediate coverage and the shortage of hydrogen supply are the two primary issues for the selective suppression of ethylene generation (*Nat. Nanotechnol.* 2023, 18, 299-306; *Nat. Catal.* 2022, 5, 251-258; *Nat. Catal.* 2020, 3, 75-82; *Nat. Catal.* 2018, 1, 764-771). However, as seen from the mechanism of CO electroreduction for various products (**Figure R1**), the bifurcation point of HOCCO* intermediate is still surrounded by protons from the HER reaction, which can be judged from the fact that hydrogen is also the product of CORR in water. Thus, the ethylene still occupies a significant portion as a main product.

Based on the above-mentioned content, "The enrichment of the COH* intermediate on copper and the suppression of HER help to decrease ethylene generation^{24,52}." is added in the revised manuscript (**Page 14, Line 259**).

Figure R1. The electroreduction reaction pathway for the formation of ethylene and oxygenates from CO (*J. Am. Chem. Soc.* 2022, 144, 20495–20506).

Comment 2. The authors assumed an Ag(100)/Cu(100) interface for the DFT and AIMD calculations and described their findings based on this assumption. However, when examining the XRD data in Figure 2a, it is evident that the Cu/Ag catalyst does not exhibit the crystallographic orientation of Ag(100) or Cu(100), but rather prominently displays Cu(111) and Ag(111) orientations. Therefore, in order to establish a connection between the DFT and AIMD calculation data and the experimental data, it is necessary to provide calculations for the Ag(111)/Cu(111) interface

Answer: The (100) crystal face is chosen for its high activity for CO₂RR (*Nat. Commun.* 2023, 14, 2387; *Nat. Catal.* 2020, 3, 98-106). According to the reviewer's suggestion, we performed calculation for the Ag(111)/Cu(111) interface. As illustrated by the time evolution snapshots (**Figure R2, Figure S32 in the revised SI**), the track profile and the corresponding energy change (**Figure R3, Figure S33 in the revised SI**), the *HCO species also exhibits a spontaneous transfer from Ag atoms to Cu atoms, which shows an exothermic adsorption energy of -0.52 eV for *HCO transfer from Ag(111) to Cu(111).

“Considering that Cu(111) surface and Ag(111) faces are typically the dominant facets of

polycrystalline Cu and Ag, we have also carried out AIMD simulations for the (111) surfaces of these crystals, and the results revealed that similar to the (100) faces, *HCO is spontaneously transferred from Ag(111) to Cu(111) (Supplementary Figs. 32,33).” is added in **Page 18, Line 319 in the revised manuscript.**

Figure R2. The typical snapshots of *HCO transfer of crossing the boundary of Ag(111) and Cu(111) with the boundary density of 1/9.

Figure R3. The trajectory profile of *HCO from Ag(111) to Cu(111) and the corresponding energy change.

To reviewer 2:

Reviewer letter: In this article, the authors investigate Cu-Ag nanocomposites for CO electroreduction to oxygenate products. Their premise is that CO adsorbs more strongly in the liquid phase on Ag than on Cu, contrary to trends in the gas phase. This has been demonstrated using TPD curves in the gas phase, LSV, and stripping experiments in the liquid phase. Therefore, if a composite Cu-Ag catalyst is used, CO adsorbs on Ag, which they claim can convert to COH* and is then transported to Cu, where it undergoes C-C coupling reactions. They used sequential deposition techniques to form the desired nanocomposites with different Cu-Ag ratios. Experimental characterization using XRD, STEM, HR-TEM shows no alloy formation is observed but the Cu nanoclusters coated by Ag. They also observe lattice fringes corresponding to the 111 facets of both metals in TEM. They then performed flow cell experiments (MEA) at high current densities to show that the FE for oxygenates is indeed high at 800 mA/cm², with Cu/30Ag performing the best. The stability of the catalyst for 28 hrs shows no degradation of performance. To understand the mechanism behind the improvement, they performed DEMS and in-situ flow-cell experiments where they observed peaks corresponding to the formation of formaldehyde on Ag while the same was not observed on Cu or Cu/30 Ag. Through this, they conclude that Ag helps in the formation of formaldehyde-like species. However, they claim that on the interface even though formaldehyde like species is produced, they are consumed very fast leading to non-detection of these species. They then performed a few DFT/AIMD simulations to check the stability of COH on interfaces and the movement across it. The article makes interesting observations regarding the role of Ag, and the characterization and catalytic experiments look promising. However, the paper raises a lot of questions, especially in terms of mechanistic aspects and the DFT simulations which do not merit the direct publication of the work.

Answer: We thank the reviewer for the positive and constructive comments on our communication. To save the reviewer's valuable time, key revisions are displayed in a yellow background in the revised manuscript and Supporting Information.

Comment 1. The authors, based on previous articles (ref. 26, 27 in the paper), make an assumption that CO to COH can be rate-determining for C₂ formation instead of C-C coupling steps. Both the references that the authors cite do not definitively conclude that only CO to COH is RDS; they claim that CO to COH/CHO can be RDS for this reaction. However, the authors only consider the COH* formation as RDS, but following their logic, how do they explain the formaldehyde product detection from their DEMS and

Raman experiments on Ag? Formaldehyde will be produced from the CHO pathway, not the COH (Energ Environ Sci 2010, 3 (9), 1311–1315). COH converts to carbon or CHOH and then leads to methane formation. I feel this assumption of COH formation as RDS cannot help in understanding or explaining the experimental results.

Answer: We sincerely acknowledge the kind comment. We provide a detailed description of the progress in the mechanistic study here by others to explain why we consider the rate-determining step of CORR and the intermediate (COH) plays an important role.

(1) The rate-determining step (RDS) of CORR

For a long time before 2021, C-C coupling step is regarded as the rate-determining step (RDS) for CO₂RR because this is in line with the lack of pH dependence in the CO₂RR, as no H⁺ or OH⁻ is involved in the RDS or based on the theoretical calculations (*J. Am. Chem. Soc.* 2016, 138, 483-486; *J. Phys. Chem. Lett.* 2016, 7, 1471-1477. *J. Am. Chem. Soc.* 2017, 139, 130-136).

Figure R1. pH effect on CO₂RR. Partial current densities of individual products on RHE scale (a) and on SHE scale (b): CO₂RR (solid symbols) and CORR reduction (open symbols); (c) CVs of polycrystalline copper electrodes in both CO and Ar purged KOH electrolyte. (*ACS Catal.* 2018, 8, 7445-7454)

To confirm the RDS of CO₂RR, Jaramillo et al. (*ACS Catal.* 2018, 8, 7445-7454) provided the new

understanding to this pH effect. On the SHE scale (**Figure R1a,b**), the overpotentials for C₂₊ products in CORR are similar to that of CO₂RR, indicating that the RDS step for C₂₊ formation is pH-independent on an absolute potential scale for both of them. Besides the C–C coupling through CO dimerization was still as one of the possibilities, the authors provided the second possibility that water could be possibly involved in RDS because proton-coupled electro-chemical reactions using water would also lead to a pH-independent result, even for alkaline solution with little proton source. Thus, C-C coupling from CO dimerization and proton-electron transfer from water during the rate-limiting step are plausible explanations for the experimentally measured pH effect.

The lack of consensus on RDS for C-C formation can be attributed to a lack of experimentally derived electrokinetic data to support their claims. For example, for the above-mentioned work, the authors evaluated pH effect under the different potentials even beyond the potential of the mass transfer limitation (**Figure R1c**).

Table R1. Summary of proposed reaction schemes for C₂₊ product (A.1-A.4) formation, and corresponding Tafel slopes. (*Nat. Commun.* 2021, 12, 3264)

	Proposed reaction scheme for C ₂₊ product formation	Tafel slope	CO order at high θ_{CO}	pH dependent
A.1	$*CO + *CO + e^- \rightarrow C_2O_2^- + *$ (RDS)	118 mV dec ⁻¹	0	No
A.2	$*CO + CO_b + e^- \rightarrow *C_2O_2^-$ (RDS)	118 mV dec ⁻¹	1	No
	$*CO + H_2O + e^- \rightarrow *CO(H) + OH^-$ (RDS)			
A.3	$*CO + *CO(H) + e^- \rightarrow C_2O_2(H)^- + *$ or $*CO(H) + CO(H) \rightarrow *C_2O_2(H)_2 + *$	118 mV dec ⁻¹	0	No
	$*CO + H^+ + e^- \rightarrow *CO(H)$ (RDS)			
A.4	$*CO + *CO(H) + e^- \rightarrow C_2O_2(H)^- + *$ or $*CO(H) + *CO(H) \rightarrow *C_2O_2(H)_2 + *$	118 mV dec ⁻¹	0	Yes

Figure R2. Tafel curves for C_2^+ product formation at different electrolyte pH. The logarithms of partial current densities for ethylene, acetate, ethanol, and n-propanol are plotted in SHE scale (a) and RHE scale (b), respectively. (*Nat. Commun.* 2021, 12, 3264)

Lu et al. (*Nat. Commun.* 2021, 12, 3264) strictly controlled the applied potential below the potential of

the mass transfer limitation and analysed the Tafel slope to identify the RDS for C_{2+} products (**Table R1**). Based on the pH independent result and Tafel slope of $\sim 120 \text{ mV dec}^{-1}$ (**Figure R2**), the authors excluded **A.4** step. Furthermore, according to the result of the dependence of C_{2+} production rates on the CO partial pressure (p_{CO}), the reaction order is less than unity in the alkaline solution at 1 atm, indicating **A.2** is unlikely the RDS for C_{2+} formation (**Figure R3**). Finally, the authors concluded that although the C-C coupling step remains likely, the hydrogenation of CO with water as the proton donor, cannot be ruled out.

Figure R3. The logarithms of partial current densities for C_2H_4 formation vs. logarithms of p_{CO} . (*Nat. Commun.* 2021, 12, 3264)

Very Recently, Xu et al. investigated the RDS of CORR combined with the CO adsorption isotherms using the *in situ* surface enhanced infrared absorption spectroscopy (*Angew. Chem. Int. Ed.* 2022, 61, e202111167). Based on the p_{CO} -dependent C_{2+} products yield rates (**Figure R4**), the reaction order will be 2nd if the C-C coupling from two CO molecules. However, data from Xu et al. and Lu et al. (*Nat. Commun.* 2021, 12, 3264) showed that the reaction is NOT 2nd order. Thus, C-C from CO dimerization is not likely the RDS step for C_{2+} . Besides, the isotopic labeling experiments can provide the direct evidence that C-C yield rate will be not disrupted if the C-C coupling is the RDS step because the rate of coupling between two CO is not expected to be impacted by switching the solvent from H_2O to D_2O as it does not involve any proton transfer. **As shown from Figure R5, ethylene formation rates were reduced by a factor of ~ 3.5 when the CORR was conducted in D_2O as compared to H_2O , directly proving that the RDS involved**

hydrogen transfer, and not C-C coupling. The Tafel slopes for ethylene production in both H₂O and D₂O were determined to be ~ 120 mV dec⁻¹, indicating the reaction mechanism did not depend on the isotopic identity of the solvent. Finally, CO + H₂O → COH matches well with the equation: $j_{C_2+} = k_{2+} \cdot \theta_{CO} \cdot \theta_{H_2O}$ and explain the reaction order of 0 at 1 atm and as well as 1st reaction order dependence on θ_{CO} at low θ_{CO} . ($j_{C_2+} = k_{2+} \cdot \theta_{CO} \cdot (1 - n\theta_{CO})$, $n\theta_{CO} + \theta_{H_2O} = 1$, $p_{CO} < 0.4$ atm) (**Figure R4**). Herein, the authors explained that COH does not provide insight into the detailed structure. Noteworthy, to further validate the reaction pathway, the authors performed co-electrolysis of CO with the alkyl iodide and then drew a pathway for various products (**Figure R6**, *J. Am. Chem. Soc.* 2022, 144, 20495–20506). COH as the RDS for CORR is verified in the electrokinetic work reported by Takanae et al. (**Figure R7**, *ACS Catal.* 2023, 13, 1791–1803).

Therefore, taking into account the wealth of recent reports validating the CO + H₂O → COH mechanisms, we can conclude that the CO + H₂O → COH RDS is the most likely mechanism.

Figure R4. The logarithms of partial current densities for C₂₊ product formation vs. logarithms of p_{CO} . (*Angew. Chem. Int. Ed.* 2022, 61, e202111167)

Figure R5. Tafel curves for ethylene formed in CORR in 1.0 M NaOH contained H₂O (black trace) and D₂O (green trace) at a p_{CO} of 1.0 atm. (*Angew. Chem. Int. Ed.* 2022, 61, e202111167)

Figure R6. The electroreduction reaction pathway for the formation of ethylene and oxygenates from CO. (*J. Am. Chem. Soc.* 2022, 144, 20495- 20506)

Figure R7. Potential Reaction Pathways toward C_{2+} Products during Electrocatalytic CO Reduction. (*ACS Catal.* 2023, 13, 1791-1803)

(2) The expression form of the intermediate (COH)

Herein, the usage of the intermediate “COH” is inherited from previous reports (*Angew. Chem. Int. Ed.* 2022, 61, e202111167; *J. Am. Chem. Soc.* 2022, 144, 20495-20506; *ACS Catal.* 2023, 13, 1791-1803). And as the referee said, COH does not provide insight into the detailed structure in terms of whether hydrogen addition occurs on C or O end of the adsorbed CO on Cu.

Here, based on our theoretical calculations in the part of Mechanistic insights, CHO is the more stable form, which is also experimentally confirmed by our DEMS and Raman results. Thus, the expression of COH intermediate in our manuscript is actually consistent with the CHO that the reviewer mentioned.

Comment 2. Based on the mechanism proposed, how do authors explain the increased selectivity to oxygenate products? If all the C-C coupling does occur on the Cu surfaces, why is there an increase in oxygenate product formation? Since Cu is inherently known to produce both ethylene and ethanol.

Answer: To answer the first question: Our proposed mechanism is based on the newest thinking in the field, backed by solid data in previous studies. As known, the formation of oxygenates originates from the combination of COH with CH_x (**Figure R8.** *Nat. Catal.* 2022, 5, 878–887). The enhanced generation of oxygenates means an increased amount of the oxygenates precursor (CH_xCO) (**Figure R9,** *J. Am. Chem.*

Soc. 2022, 144, 20495–20506). According to our experimental evidences: a. Ag can provide a rich COH surrounding for Cu. b. the combination of COH with CH_x generated from Cu produces the abundant oxygenates precursor. c. the spontaneous transfer of COH has been confirmed by *ab initio* molecular dynamics (AIMD) simulations. This catalytic process can also be embodied in our Ag-dependent CORR performance, *in situ* Raman and online DEMS.

Thus, the continuous supply of CH_xCO leads to the enhanced oxygenates generation.

Figure R8. Asymmetric C-C coupling in inorganic CO₂ electroreduction. (*Nat. Catal.* 2022, 5, 878–887).

Figure R9. Various oxygenates formation from CH_xCO. (*J. Am. Chem. Soc.* 2022, 144, 20495–20506).

(1) If all the C-C coupling does occur on the Cu surfaces, why is there an increase in oxygenate product formation? Since Cu is inherently known to produce both ethylene and ethanol.

First, we have shown pure Ag performance for oxygenates generation at the optimized condition of 800 cm mA⁻², as shown in **Figure R10** and **Table R2** (**Figure S13** and **Table S6** in the revised SI). The very weak performance for oxygenates electrosynthesis over pure Ag indicates the primary reaction zone is

located on Cu. Compared with pure Cu, the increase in selectivity of Cu in Cu/30Ag for oxygenates generation can be attributed to the increased coverage of oxygenates precursor in the presence of Ag (enhanced CO capture) and the shortage of hydrogen supply. These two factors can suppress the ethylene generation, and at the same time increase the oxygenates generation (*Nat. Nanotechnol.* 2023, 18, 299-306; *Nat. Catal.* 2022, 5, 251-258; *Nat. Catal.* 2020, 3, 75-82; *Nat. Catal.* 2018, 1, 764-771).

Based on the above-mentioned content, "The enrichment of the COH* intermediate on copper and the suppression of HER help to decrease ethylene generation^{24,52}." is added in the revised manuscript (Page 14, Line 259).

Figure R10. The performance of oxygenates electrosynthesis over Cu/30Ag and Ag at the applied current density of 800 mA cm⁻².

Table R2. The CO-EC performance of Ag for oxygenates generation at 800 mA cm⁻² (unit: % for FE, mA cm⁻² for *j*).

Products	800 mA cm ⁻²	
	FE and j	
Acetate	1.17 ± 0.02	
	9.36 ± 0.16	
Acetaldehyde	0.06 ± 0.01	
	0.48 ± 0.08	
Propionaldehyde	0	
	0	
Acetone	0.02 ± 0.00	
	0.16 ± 0.00	
Ethanol	0.38 ± 0.02	
	3.04 ± 0.16	
n-propanol	0.03 ± 0.00	
	0.24 ± 0.00	
Ally Alcohol	0.01 ± 0.00	
	0.08 ± 0.00	
1-butanol	0	
	0	
Methanol	0	
	0	
Oxygenates	1.66 ± 0.05	
	13.28 ± 0.40	

Comment 3. The authors should also consider going through the following paper and possibly cite it: ACS Catal 2020, 10 (7), 4059–4069. Even in this work, CuAg composite catalysts were used albeit for CO₂ reduction with more detailed DFT simulations.

Answer: According to the reviewer's nice suggestion, this work of Yeo et al. has been read and cited (**Ref. 9**). In this work, the authors claimed CO dimers was circumvented on Cu and a novel pathway for ethanol formation was proposed. If more *in situ* experimental techniques can be carried out, the statement will be more solid.

Comment 4. Furthermore, the current set of DFT simulations in the paper adds minimal value to the study while a lot more can be done.

Answer: Based on the reviewer's questions, more DFT simulations are calculated and shown in the revised manuscript and SI.

Comment 4a. Firstly, the basic assumption of using Cu(100)|Ag(100) interface is not at all justified. The HRTEM in Figure 2(d) shows lattice fringes corresponding to Cu(111)|Ag(111). Why wasn't this interface simulated? The current Cu(100)|Ag(100) interface looks very sub-optimal to me; what is the lattice strain of the interfaces when they are placed side-to-side.

Answer: We sincerely acknowledge the kind comment. The (100) crystal face is chosen for its high activity for CO₂RR (*Nat. Commun.* 2023, 14, 2387; *Nat. Catal.* 2020, 3, 98-106). According to the reviewer's suggestion, we have also performed calculations on the Ag(111)/Cu(111) interface. As illustrated by the snapshots (**Figure R11, Figure S32 in the revised SI**), the track profile and the corresponding energy change (**Figure R12, Figure S33 in the revised SI**), the *HCO species also exhibits a spontaneous transfer phenomenon from Ag atoms to Cu atoms, which shows an exothermic adsorption energy of -0.52 eV for *HCO transfer from Ag(111) to Cu(111).

"Considering that Cu(111) surface and Ag(111) faces are typically the dominant facets of polycrystalline Cu and Ag, we have also carried out AIMD simulations for the (111) surfaces of these crystals, and the results revealed that similar to the (100) faces, *HCO is spontaneously transferred from Ag(111) to Cu(111) (Supplementary Figs. 32,33)." The revised statement is added in **Page 18, Line 319 in the revised manuscript**.

Figure R11. The typical snapshots of *HCO transfer of crossing the boundary of Ag(111) and Cu(111) with the boundary density of 1/9.

Figure R12. The trajectory profile of *HCO from Ag(111) to Cu(111) and the corresponding energy change.

Given that the lattice constant of Ag surpasses that of Cu, there is anticipated extension or stress near the boundary region. To mitigate this, the Cu(100)|Ag(100) interface was constructed based on the fully relaxed bulk Cu(100)|Ag(100) with an identical atomic arrangement, aiming to minimize extension or stress, particularly in the boundary area. The calculated lattice constants for pure Ag and Cu are 4.126 Å and 3.615 Å, respectively. Consequently, the Cu(100)|Ag(100) interface model adopts modified lattice constants of 3.910 Å and 4.086 Å along the x and y directions, respectively.

In the Cu(100)|Ag(100) interface model, a general compression is observed in the Ag phase, while the Cu phase experiences stretching, especially in the bottom slabs. However, on the top layer of the interface model, the strain effect is minimized due to an ample space for Ag and Cu atoms to adjust to their most suitable lattice, resulting in the characteristic blue bending curve of atoms at the interface, as illustrated in the **Figure R13**.

Figure R13. The interface structure of Cu(100)/Ag(100).

Comment 4b. What is the difference in thermodynamic or kinetic barriers for COH/CHO formation from CO on Ag vs. Cu surfaces. Since CO hydrogenation step is claimed to be the RDS in this study and how does it compare to C-C coupling reaction energetics.

Answer: According to the reviewer's suggestion, the simulations encompassing CO hydrogenation and CO dimerization reactions are carried out. Initially, the CO hydrogenation pathway leads to the formation of *HCO instead of *COH. As detailed in **Table R3 (Table S14 in the revised SI)**, the disparity in adsorption Gibbs energy between *HCO and *COH ranges from -0.13 eV to -0.86 eV. Notably, the

adsorption energy differences between Ag(100) and Cu(100), as well as Ag(111) and Cu(111), are -1.02 eV and -0.52 eV, respectively. These differences serve as the thermodynamic driving forces facilitating the transfer of *HCO from Ag to Cu. As indicated in the subsequent **Table R3 (Table S14 in the revised SI)**, the CO hydrogenation steps on Ag and Cu surfaces exhibit energy ranges from 0.45 eV to 0.69 eV. Notably, the generation of *HCO can be influenced by applied potentials, altering the Gibbs free energy by -eU, as outlined in the Computational Hydrogen Electrode (CHE) model proposed by Nørskov (*J. Phys. Chem. B* 2004, 108, 17886-17892). The corresponding change in ΔG is illustrated in **Figure R14 (Figure S30 in the revised SI)**, taking Ag(100) as an example. This analysis underscores that *CO hydrogenation step can thermodynamically transpire at negative potential below -0.45 V on Ag(100).

However, the CO dimerization reactions are thermodynamically implausible across all surfaces, as indicated by the Gibbs free energy changes ranging from 0.86 eV to 1.50 eV. According to Calle-Vallejo and Koper's work (*Angew. Chem. Int. Ed.* 2013, 52, 7282-7285), CO dimerization step is promoted by the negatively charged CO dimer, i.e., $C_2O_2^-$, the Gibbs free energy of which is further modulated by -eU. According to this framework, the thermodynamic feasibility of the CO dimerization step requires a minimum of -0.86 V. As a result, on Ag or Cu surfaces, the prevailing tendency for CO is to undergo hydrogenation to form *HCO rather than dimerization.

Table R3. The Gibbs free energy change of the CO hydrogenation and CO dimerization reactions on different surfaces. The reference moleculars are CO and H₂ in the vacuum.

	Ag(100)	Ag(111)	Cu(100)	Cu(111)
$G_{ad,*HCO} - G_{ad,*COH} / eV$	-0.67	-0.86	-0.67	-0.13
$G_{ad,*HCO} / eV$	0.68	0.34	-0.34	-0.18
ΔG of CO hydrogenation / eV	0.45	0.47	0.56	0.69
ΔG of CO dimerization / eV	0.86	1.15	1.07	1.50

Figure R14. The Gibbs free energy change of CO hydrogenation with the change of the applied potentials on Ag(100).

Comment 4c. The authors make an interesting claim through experiments in Figure 1 that CO adsorption can actually be stronger on Ag as compared to Cu in a liquid medium. Can this be proved using DFT simulations?

Answer: We thank the reviewer for this valuable comment.

1. The influence of the solvent on CO adsorption over Cu and Ag

The CO adsorption in a liquid medium was simulated with the employment of the implicit solvation model (*Phys. Rev. Lett.* 1996, 77, 3865-3868.) and a relative permittivity of 80 set for simulating the aqueous electrolyte. However, the modest solvation effect from water only adjusts the adsorption energy within a narrow range of 0.10 eV. This adjustment does not contribute significantly to enhancing CO binding on Ag compared to Cu.

2. CO adsorption on Cu and Ag in gas phase

CO adsorption on pure metals in gas has been extensively studied with the general conclusion that CO adsorbs stronger on Cu than Ag (*Phys. Rev. B* 2019, 100, 035442, *Mater. Today Commun.* 2020, 24, 101158). We have the same conclusion that CO adsorbs stronger on Cu than Ag in vacuum (E_{ad} is -1.11

eV and -0.42 eV, respectively, lower on Cu(100) and Cu(111) than that on Ag(100) and Cu(111)).

3. The change of CO dipole moment on different metals and the corresponding CO adsorption behaviors in the presence of the electric field

Figure R15. The adsorption of CO on Ag(100) (left) and Cu(100) (right) with charge distribution and Bader charge analysis.

Taking *CO adsorption on Ag(100) and Cu(100) for example, the C and O atoms on Ag(100) exhibit an additional charge of -1.638e and 1.836e, respectively (**Figure R15**). On the other hand, on Cu(100), these values are -1.428e and 1.778e, respectively. Consequently, Ag and Cu surfaces contribute 0.198e and 0.35e, respectively, to *CO adsorbates. The interaction between *CO and metal surfaces enhances adsorption while simultaneously weakening the C≡O triple bonds. This effect is more pronounced on Cu(100), leading to stronger adsorption of *CO and longer C-O bond distance.

However, the charge separation between *CO results in a dipole moment, distinct from that in the gaseous CO molecule. The dipole moment ($\mu=qd$) of *CO on Ag(100) (2.1459) is slightly higher than that on Cu(100) (2.0863). This suggests that *CO on Ag(100) could be more stretched, and *CO adsorption might be more enhanced than on Cu(100) under the same external electric field. As indicated by Rossmeisl et al. (*J. Phys. Chem. B* 2006, 110, 21833-21839), the energy of the system under external electric field could be expressed as:

$$\Delta E(\epsilon) = \mu\epsilon - \frac{1}{2}\alpha\epsilon^2 + \dots$$

Where ϵ is the electric field, μ is the dipole moment in the direction of the electric field, α is the static

polarizability. Due to the inversion symmetry inherent in our metal slab models, the dipole moment (μ) is zero. It is noteworthy that Ag exhibits a higher polarizability than Cu, as evident from the steeper sharpness in **Figure R16**. Consequently, electrons on Ag surfaces flow more readily to *CO than on Cu surfaces. This ultimately gives rise to the potential scenario where *CO adsorbs more strongly on Ag than on Cu under an external electric field. The results depicted in **Figure R17 (Figure S4 in the revised SI)** indeed substantiate the assertion that CO adsorption is more robust on Ag than on Cu, particularly with an applied electric force field exceeding ~ 1.2 eV/Å.

Figure R16. The energy change of the metal slabs as a function of the external electric field.

Figure R17. The electric force field effect on CO adsorbate

Consequently, the imposition of an electric field can lead to a tighter binding of CO on Ag compared to Cu.

“DFT calculations verify that the presence of an electric field is key to the capture of CO by Ag in liquid phase (Supplementary Fig. 4)” is added in **Page 6, Line 120 in the revised manuscript**.

Comment 4d. In the AIMD simulations in Figure 5c, how can authors justify a change in energy of 6 eV? Furthermore, I cannot fathom COH* displacing so much and crossing the interface in just 2 ps of AIMD simulations. I would have expected the authors to use some accelerated AIMD techniques like meta dynamics or slow-growth simulations to simulate this process.

Answer: We sincerely acknowledge the kind comment. The observed 6 eV energy change in the system arises from the construction of the interface model. The liquid-solid model adopts an arrangement of water molecules resembling ice-like crystal structures due to the challenges in replicating the practical disorder of the liquid water phase. Subsequently, geometry optimization with DFT is employed to attain a structure closely resembling its equilibrium state. However, in molecular dynamic simulations, the initial water structure at the simulated temperature of 298.15 K deviates from the equilibrium state of liquid water, resulting in the observed total energy change of 6 eV from the initial state to the equilibrium state. Notably, the energy undergoes a rapid decline in the initial tens of fs, indicative of the

pre-equilibrium stage. Once the system reaches a local equilibrium state, as illustrated in **Figure 5c in the revised manuscript** stage 1, the energy stabilizes significantly compared to the pre-equilibrium stage. The transfer of *HCO from Ag(100) to Cu(100) in the simulated system experiences an average energy decrease of approximately 1.3 eV, closely aligning with the theoretical thermodynamic driving force of -1.02 eV.

Regarding the 2 ps timescale for the *HCO transfer, AIMD simulations were following the typical Born–Oppenheimer molecular dynamics (BOMD) method in that the quantum mechanical effect of the electrons is included in the calculation of energy and forces for the classical motion of the nuclei. Accelerated AIMD techniques were not employed. The relatively short duration of 2 ps for *HCO transfer is rooted in the following reasons. Firstly, *HCO on Ag(100) holds the dynamic nature of the motion. *HCO is weakly adsorbed on pure Ag(100) surface with the adsorption energy of 0.68 eV, indicating the weak interaction between *HCO and the surface. The solvation effect from water by forming hydrogen bonds will further benefit *HCO movement with the dynamics of water (in snapshot pictures, the water molecules blocking *HCO were deleted and H-bond could disappear). Secondly, the adsorption energy of *HCO on different sites of Ag(100), including top, bridge, and hollow positions, is closely matched, with a maximum difference of 0.07 eV (**Figure R18**), which hints low barriers for *HCO transferring from one adsorption site to another. Thirdly, the quick transfer of *HCO presented in **Figure 5c in the revised manuscript** is also facilitated by the fact that *HCO was initially adsorbed on the boundary Ag atom where the thermodynamic driving force is the maximum in the interface model, which quickly promotes its transfer. When *HCO adsorbs on Ag(100) terrace atom, as shown in **Figure R19**, it does not show the immediate transfer phenomenon, but jumps from one Ag atom to another and finally reaches the boundary. **Figure R19** also reveals the rapid movement of *HCO between adsorption sites, from Ag atom to the neighboring one could happen in less than 1 ps. All these facts point to the dynamic transfer of *HCO from Ag atom to Ag-Cu boundary and Cu atom within 2 ps.

Figure R18. *HCO on different sites of Ag(100).

a

Figure R19. The typical snapshots of *HCO transfer (a), the trajectory profile of *HCO and the corresponding energy change on Ag(100) (b).

To reviewer 3:

Reviewer letter: Herein, Meng et al successfully address a long-standing scholarly debate concerning CO capture and activation over Ag. Furthermore, leveraging this property, they propose a rationally designed strategy to employ Ag as a catalytic site for capturing CO and converting it into a COH* oxygenates precursor. This process creates a COH*-rich environment around Cu nanoclusters, serving as a highly efficient relay electrocatalyst for synthesizing oxygenates products. The effectiveness of Ag and the transfer of COH* from Ag to Cu are solidly demonstrated through CO stripping, in situ DEMS/Raman/XAFS experiments, robustly supported by Ab initio molecular dynamics simulations. By briefly controlling the areal density of Ag-Cu interfaces to match or balance the generation rate of the oxygenates precursor with its consumption rate, the record partial current density of 800 mA cm⁻² with decent 67% selectivity in 1200 mA cm⁻² CO electrolysis is reached. Significantly, through optimization of reaction parameters, this results in an approximate profit of 234 USD per tonne of acetic acid. As a result, this well-organized manuscript offers a novel understanding of the Cu-Ag catalytic system and contributes to enhancing overall electrolysis efficiency for oxygenates electrolysis. Given the significance of these findings and the solid experimental/mechanism analysis, I strongly recommend the publication of this manuscript in Nature Communications. Minor revisions are suggested to address the following issues.

Answer: We thank the reviewer for the positive and constructive comments on our communication. To save the reviewer's valuable time, key revisions are displayed in a yellow background in the revised manuscript and Supporting Information.

Comment 1. For further reflecting the CO capture capacity of Ag in the composite, the CO stripping curve of the champion Cu/30Ag catalyst should be added.

Answer: According to the reviewer's suggestion, the CO stripping curve of the champion Cu/30Ag catalyst is measured under the CO atmosphere using that under the Ar atmosphere as the comparison and the result is shown in **Figure R1** and **Figure S3 in the revised SI**. As shown in **Figure R1**, compared with the CO stripping peak located at 0.27 V over Cu, the peak position shifts to 0.31 V on Cu/30Ag, indicating Ag can efficiently enhance the CO capture in the solution.

Figure R1. The stripping curve over Cu/30Ag in CO (a) and Ar (b) atmospheres.

Comment 2. The $U-t$ curves over the champion catalyst under the different applied current densities (50 mA cm^{-2} to 800 mA cm^{-2}) should be provided for further describing its stability.

Answer: Thanks for the comment. The $U-t$ curves over the champion catalyst under the different applied current densities from 50 mA cm^{-2} to 800 mA cm^{-2} are provided. As seen from **Figure R2** and **Figure S12 in the revised SI**, the curves of $U-t$ under the different applied current densities show a stable state, indicating the excellent catalytic stability of Cu/30Ag.

Figure R2. The typical $U-t$ curves over Cu/30Ag under the different applied current densities for CO electrolysis.

Comment 3. Table of Contents (TOC) of this work can be drawn and provided.

Answer: We sincerely acknowledge the kind comment. According to the editor's request, we do not show TOC of this work in our revised manuscript at this stage.

Comment 4. To enhance the reproducibility of this study, the additional information regarding the electrode preparation process should be added.

Answer: According to the reviewer's nice suggestion, the electrode preparation process is detailed described and shown in the revised manuscript (**Page 20, line 366**) as follows:

Specifically, the catalyst ink was prepared by adding 100 μL of Nafion into 5 mL of isopropanol in which 25 mg of catalyst powder was dispersed, followed by ultrasonication for 30 min. The well-mixed catalyst ink was then spray-coated onto a GDE (5 cm \times 5 cm) by airbrushing to achieve a catalyst loading of $\sim 1 \text{ mg cm}^{-2}$.

Comment 5. Page 10, line 22. '234 USD per tonne of the products' should be revised to '234 USD per

tonne of the product' or '234 USD per tonne of the acetate acid product'

Answer: We sincerely acknowledge the kind comment. '234 USD per tonne of the products' has been revised to '234 USD per tonne of the acetate acid product' (**Page 10, Line 187**).

REVIEWERS' COMMENTS

Reviewer #2 (Remarks to the Author):

Authors did a commendable job in clarifying and answering the reviewer questions especially in detailing the mechanistic insights. However not all the details shown in the answer to reviewer questions have been added to the manuscript/SI.

1.No corresponding text for Figures S13, S30 and Tables S6, S14 has been added in the revised manuscript.

2.I would recommend adding the lattice strain percentages of the interface as compared to pure metal surfaces in the SI or methods section.

After addressing these comments, I would recommend this very interesting manuscript for publication.

A point-by-point response to the reviewer's comments

To reviewer 2:

Reviewer letter: Authors did a commendable job in clarifying and answering the reviewer questions especially in detailing the mechanistic insights. However not all the details shown in the answer to reviewer questions have been added to the manuscript/Sl.

Answer: We highly appreciate the reviewer for the positive and constructive comments on our communication. To save the reviewer's valuable time, key revisions are displayed in yellow background in the revised manuscript.

Comment 1. No corresponding text for Figures S13, S30 and Tables S6, S14 has been added in the revised manuscript.

Answer: We sincerely acknowledge the kind comment. According to the reviewer's suggestion, we have now revised.

1. "while pure Ag shows around 1 % FE for oxygenates" is added for describing **Supplementary Figure 13** and **Table 6 (Page 8, line 169 in the revised manuscript)**.
2. "These results are also appropriate for between Ag (111) and Cu(111) and notably theoretical analysis indicates that hydrogenation of CO is thermodynamically more feasible than CO dimerization on both Ag and Cu surfaces (**Supplementary Figure 30 and Table 14**)" is added for describing **Supplementary Figure 30 and Table 14 (Page 13, line 275 in the revised manuscript)**.

Comment 2. I would recommend adding the lattice strain percentages of the interface as compared to pure metal surfaces in the SI or methods section.

Answer: We sincerely acknowledge the kind comment. According to the reviewer's suggestion, we have now revised.

"Given that the lattice constant of Ag surpasses that of Cu, there is anticipated extension or stress near the boundary region. To mitigate this, the Cu(100)|Ag(100) interface was constructed based on the fully relaxed bulk Cu(100)|Ag(100) with an identical atomic arrangement, aiming to minimize extension or

stress, particularly in the boundary area. The calculated lattice constants for pure Ag and Cu are 4.126 Å and 3.615 Å, respectively. Consequently, the Cu(100)|Ag(100) interface model adopts modified lattice constants of 3.910 Å and 4.086 Å along the x and y directions, respectively. This corresponds to the stretching strain of 8.16 % and 13.02 % along x and y direction, respectively, compared to pure Cu, as well as the compressing strain of 5.23 % and 0.97 % along x and y, direction, respectively, compared to pure Ag. However, on the top layer of the interface model, the strain effect is minimized due to an ample space for Ag and Cu atoms to adjust to their most suitable lattice, resulting in the characteristic blue bending curve of atoms at the interface, as illustrated in the **Supplementary Figure 34**” is added in **SI**, corresponding to **Page20, line 431 in the revised manuscript**.

We acknowledge all the kind comments and wise suggestions from the reviewer.

We are sure that the quality of this work will be greatly improved according to these helpful comments and wise suggestions.